# A Graph-Theoretic Framework for Understanding Open-World Semi-Supervised Learning

**Yiyou Sun, Zhenmei Shi, Yixuan Li**
Department of Computer Sciences
University of Wisconsin, Madison
`{sunyiyou,zhmeishi,sharonli}@cs.wisc.edu`

## Abstract

Open-world semi-supervised learning aims at inferring both known and novel classes in unlabeled data, by harnessing prior knowledge from a labeled set with known classes. Despite its importance, there is a lack of theoretical foundations for this problem. This paper bridges the gap by formalizing a graph-theoretic framework tailored for the open-world setting, where the clustering can be theoretically characterized by graph factorization. Our graph-theoretic framework illuminates practical algorithms and provides guarantees. In particular, based on our graph formulation, we apply the algorithm called Spectral Open-world Representation Learning (SORL), and show that minimizing our loss is equivalent to performing spectral decomposition on the graph. Such equivalence allows us to derive a provable error bound on the clustering performance for both known and novel classes, and analyze rigorously when labeled data helps. Empirically, SORL can match or outperform several strong baselines on common benchmark datasets, which is appealing for practical usage while enjoying theoretical guarantees. Our code is available at `https://github.com/deeplearning-wisc/sorl`.

## 1 Introduction

Machine learning models in the open world inevitably encounter data from both known and novel classes [2, 15, 16, 65, 79]. Traditional supervised machine learning models are trained on a closed set of labels, and thus can struggle to effectively cluster new semantic concepts. On the other hand, open-world semi-supervised learning approaches, such as those discussed in studies [7, 63, 69], enable models to distinguish *both known and novel classes*, making them highly desirable for real-world scenarios. As shown in Figure 1, the learner has access to a labeled training dataset $\mathcal{D}_l$ (from known classes) as well as a large unlabeled dataset $\mathcal{D}_u$ (from both known and novel classes). By optimizing feature representations jointly from both labeled and unlabeled data, the learner aims to create meaningful cluster structures that correspond to either known or novel classes. With the explosive growth of data generated in various

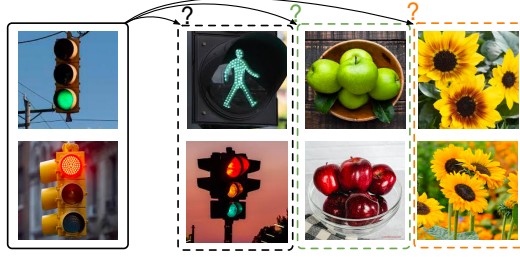

Labeled Data $\mathcal{D}_l$      Unlabeled Data $\mathcal{D}_u$

Figure 1: Open-world Semi-supervised Learning aims to correctly cluster samples in the novel class and classify samples in the known classes by utilizing knowledge from the labeled data. An open question is *"what is the role of the label information in shaping representations for both known and novel classes?"* This paper aims to provide a formal understanding.

37th Conference on Neural Information Processing Systems (NeurIPS 2023).

domains, open-world semi-supervised learning has emerged as a crucial problem in the field of machine learning.

**Motivation.** Different from self-supervised learning [5, 8, 11, 12, 23, 26, 68, 77], open-world semi-supervised learning allows harnessing the power of the labeled data for possible knowledge sharing and transfer to unlabeled data, and from known classes to novel classes. In this joint learning process, we argue that interesting intricacies can arise—the labeled data provided may be beneficial or unhelpful to the resulting clusters. We exemplify the nuances in Figure 1. In one scenario, when the model learns the labeled known classes (e.g., traffic light) by pushing red and green lights closer, such a relationship might transfer to help cluster green and red apples into a coherent cluster. Alternatively, when the connection between the labeled data and the novel class (e.g., flower) is weak, the benefits might be negligible. We argue—perhaps obviously—that a formalized understanding of the intricate phenomenon is needed.

**Theoretical significance.** To date, theoretical understanding of open-world semi-supervised learning is still in its infancy. In this paper, we aim to fill the critical blank by analyzing this important learning problem from a rigorous theoretical standpoint. Our exposition gravitates around the open question: *what is the role of labeled data in shaping representations for both known and novel classes?* To answer this question, we formalize a graph-theoretic framework tailored for the open-world setting, where the vertices are all the data points and connected sub-graphs form classes (either known or novel). The edges are defined by a combination of supervised and self-supervised signals, which reflects the availability of both labeled and unlabeled data. Importantly, this graph facilitates the understanding of open-world semi-supervised learning from a spectral analysis perspective, where the clustering can be theoretically characterized by graph factorization. Based on the graph-theoretic formulation, we derive a formal error bound by contrasting the clustering performance for all classes, before and after adding the labeling information. Our Theorem 4.2 reveals the sufficient condition for the improved clustering performance for a class. Under the K-means measurement, the unlabeled samples in one class can be better clustered, if their overall connection to the labeled data is stronger than their self-clusterability.

**Practical significance.** Our graph-theoretic framework also illuminates practical algorithms with provided guarantees. In particular, based on our graph formulation, we present the algorithm called Spectral Open-world Representation Learning (SORL) adapted from Sun et al. [64]. Minimizing this loss is equivalent to performing spectral decomposition on the graph (Section 3.2), which brings two key benefits: (1) it allows us to analyze the representation space and resulting clustering performance in closed-form; (2) practically, it enables end-to-end training in the context of deep networks. We show that our learning algorithm leads to strong empirical performance while enjoying theoretical guarantees. The learning objective can be effectively optimized using stochastic gradient descent on modern neural network architecture, making it desirable for real-world applications.

## 2  Problem Setup

We formally describe the data setup and learning goal of open-world semi-supervised learning [7].

**Data setup.** We consider the empirical training set $\mathcal{D}_l \cup \mathcal{D}_u$ as a union of labeled and unlabeled data.

1. The labeled set $\mathcal{D}_l = \{\bar{x}_i, y_i\}_{i=1}^n$, with $y_i \in \mathcal{Y}_l$. The label set $\mathcal{Y}_l$ is known.

2. The unlabeled set $\mathcal{D}_u = \{\bar{x}_i\}_{i=1}^m$, where each sample $\bar{x}_i$ can come from either known or novel classes[1]. Note that we do not have access to the labels in $\mathcal{D}_u$. For mathematical convenience, we denote the underlying label set as $\mathcal{Y}_{\text{all}}$, where $\mathcal{Y}_l \subset \mathcal{Y}_{\text{all}}$. We denote $C = |\mathcal{Y}_{\text{all}}|$ the total number of classes.

The data setup has practical value for real-world applications. For example, the labeled set is common in supervised learning; and the unlabeled set can be gathered for free from the model's operating environment or the internet. We use $\mathcal{P}_l$ and $\mathcal{P}$ to denote the marginal distributions of labeled data and all data in the input space, respectively. Further, we let $\mathcal{P}_{l_i}$ denote the distribution of labeled samples with class label $i \in \mathcal{Y}_l$.

---

[1]This generalizes the problem of Novel Class Discovery [19, 22, 27, 28, 82, 83], which assumes the unlabeled set is *purely* from novel classes.

**Learning target.** Under the setting, our goal is to learn distinguishable representations *for both known and novel classes* simultaneously. The representation quality will be measured using classic metrics, such as K-means clustering accuracy, which we will define mathematically in Section 4.2.2. Unlike classic semi-supervised learning [86], we place no assumption on the unlabeled data and allow its semantic space to cover both known and novel classes. The problem is also referred to as open-world representation learning [63], which emphasizes the role of good representation in distinguishing both known and novel classes.

**Theoretical analysis goal.** We aim to comprehend the role of label information in shaping representations for both known and novel classes. It's important to note that our theoretical approach aims to understand the perturbation in the clustering performance by labeling existing, previously unlabeled data points within the dataset. By contrasting the clustering performance before and after labeling these instances, we uncover the underlying structure and relations that the labels may reveal. This analysis provides invaluable insights into how labeling information can be effectively leveraged to enhance the representations of both known and novel classes.

# 3 A Spectral Approach for Open-world Semi-Supervised Learning

In this section, we formalize and tackle the open-world semi-supervised learning problem from a graph-theoretic view. Our fundamental idea is to formulate it as a clustering problem—where similar data points are grouped into the same cluster, by way of possibly utilizing helpful information from the labeled data $\mathcal{D}_l$. This clustering process can be modeled by a graph, where the vertices are all the data points and classes form connected sub-graphs. Specifically, utilizing our graph formulation, we present the algorithm — Spectral Open-world Representation Learning (SORL) in Section 3.2. The process of minimizing the corresponding loss is fundamentally analogous to executing a spectral decomposition on the graph.

## 3.1 A Graph-Theoretic Formulation

We start by formally defining the augmentation graph and adjacency matrix. For clarity, we use $\bar{x}$ to indicate the natural sample (raw inputs without augmentation). Given an $\bar{x}$, we use $\mathcal{T}(x|\bar{x})$ to denote the probability of $x$ being augmented from $\bar{x}$. For instance, when $\bar{x}$ represents an image, $\mathcal{T}(\cdot|\bar{x})$ can be the distribution of common augmentations [11] such as Gaussian blur, color distortion, and random cropping. The augmentation allows us to define a general population space $\mathcal{X}$, which contains all the original images along with their augmentations. In our case, $\mathcal{X}$ is composed of augmented samples from both labeled and unlabeled data, with cardinality $|\mathcal{X}| = N$. We further denote $\mathcal{X}_l$ as the set of samples (along with augmentations) from the labeled data part.

We define the graph $G(\mathcal{X}, w)$ with vertex set $\mathcal{X}$ and edge weights $w$. To define edge weights $w$, we decompose the graph connectivity into two components: (1) self-supervised connectivity $w^{(u)}$ by treating all points in $\mathcal{X}$ as entirely unlabeled, and (2) supervised connectivity $w^{(l)}$ by adding labeled information from $\mathcal{P}_l$ to the graph. We proceed to define these two cases separately.

First, by assuming all points as unlabeled, two samples $(x, x^+)$ are considered a **positive pair** if:

> **Unlabeled Case (u):** *$x$ and $x^+$ are augmented from the same image $\bar{x} \sim \mathcal{P}$.*

For any two augmented data $x, x' \in \mathcal{X}$, $w_{xx'}^{(u)}$ denotes the marginal probability of generating the pair:

$$w_{xx'}^{(u)} \triangleq \mathbb{E}_{\bar{x} \sim \mathcal{P}} \mathcal{T}(x|\bar{x}) \mathcal{T}(x'|\bar{x}) , \tag{1}$$

which can be viewed as self-supervised connectivity [11, 23]. However, different from self-supervised learning, we have access to the labeled information for a subset of nodes, which *allows adding additional connectivity to the graph*. Accordingly, the positive pair can be defined as:

> **Labeled Case (l):** *$x$ and $x^+$ are augmented from two labeled samples $\bar{x}_l$ and $\bar{x}'_l$ with the same known class $i$. In other words, both $\bar{x}_l$ and $\bar{x}'_l$ are drawn independently from $\mathcal{P}_{l_i}$.*

Considering both case (u) and case (l), the overall edge weight for any pair of data $(x, x')$ is given by:

$$w_{xx'} = \eta_u w_{xx'}^{(u)} + \eta_l w_{xx'}^{(l)}, \text{ where } w_{xx'}^{(l)} \triangleq \sum_{i \in \mathcal{Y}_l} \mathbb{E}_{\bar{x}_l \sim \mathcal{P}_{l_i}} \mathbb{E}_{\bar{x}'_l \sim \mathcal{P}_{l_i}} \mathcal{T}(x|\bar{x}_l) \mathcal{T}(x'|\bar{x}'_l) , \tag{2}$$

and $\eta_u, \eta_l$ modulates the importance between the two cases. The magnitude of $w_{xx'}$ indicates the "positiveness" or similarity between $x$ and $x'$. We then use $w_x = \sum_{x' \in \mathcal{X}} w_{xx'}$ to denote the total edge weights connected to a vertex $x$.

**Remark: A graph perturbation view.** With the graph connectivity defined above, we can now define the adjacency matrix $A \in \mathbb{R}^{N \times N}$ with entries $A_{xx'} = w_{xx'}$. Importantly, the adjacency matrix can be decomposed into two parts:

$$A = \eta_u A^{(u)} + \underset{\downarrow \textit{Perturbation by adding labels}}{\eta_l A^{(l)}}, \tag{3}$$

which can be regarded as the self-supervised adjacency matrix $A^{(u)}$ perturbed by additional labeling information encoded in $A^{(l)}$. This graph perturbation view serves as a critical foundation for our theoretical analysis of the clustering performance in Section 4. As a standard technique in graph theory [14], we use the *normalized adjacency matrix* of $G(\mathcal{X}, w)$:

$$\tilde{A} \triangleq D^{-\frac{1}{2}} A D^{-\frac{1}{2}}, \tag{4}$$

where $D \in \mathbb{R}^{N \times N}$ is a diagonal matrix with $D_{xx} = w_x$. The normalization balances the degree of each node, reducing the influence of vertices with very large degrees. The normalized adjacency matrix defines the probability of $x$ and $x'$ being considered as the positive pair from the perspective of augmentation, which helps derive the learning loss as we show next.

## 3.2 SORL: Spectral Open-World Representation Learning

We present an algorithm called Spectral Open-world Representation Learning (SORL), which can be derived from a spectral decomposition of $\tilde{A}$. The algorithm has both practical and theoretical values. First, it enables efficient end-to-end training in the context of modern neural networks. More importantly, it allows drawing a theoretical equivalence between learned representations and the top-$k$ singular vectors of $\tilde{A}$. Such equivalence facilitates theoretical understanding of the clustering structure encoded in $\tilde{A}$. Specifically, we consider low-rank matrix approximation:

$$\min_{F \in \mathbb{R}^{N \times k}} \mathcal{L}_{\mathrm{mf}}(F, A) \triangleq \left\| \tilde{A} - FF^\top \right\|_F^2 \tag{5}$$

According to the Eckart–Young–Mirsky theorem [17], the minimizer of this loss function is $F_k \in \mathbb{R}^{N \times k}$ such that $F_k F_k^\top$ contains the top-$k$ components of $\tilde{A}$'s SVD decomposition.

Now, if we view each row $\mathbf{f}_x^\top$ of $F$ as a scaled version of learned feature embedding $f : \mathcal{X} \mapsto \mathbb{R}^k$, the $\mathcal{L}_{\mathrm{mf}}(F, A)$ can be written as a form of the contrastive learning objective. We formalize this connection in Theorem 3.1 below[2].

**Theorem 3.1.** *We define $\mathbf{f}_x = \sqrt{w_x} f(x)$ for some function $f$. Recall $\eta_u, \eta_l$ are coefficients defined in Eq.* (2). *Then minimizing the loss function $\mathcal{L}_{\mathrm{mf}}(F, A)$ is equivalent to minimizing the following loss function for $f$, which we term* ***Spectral Open-world Representation Learning (SORL):***

$$\mathcal{L}_{SORL}(f) \triangleq -2\eta_l \mathcal{L}_1(f) - 2\eta_u \mathcal{L}_2(f) + \eta_l^2 \mathcal{L}_3(f) + 2\eta_l \eta_u \mathcal{L}_4(f) + \eta_u^2 \mathcal{L}_5(f), \tag{6}$$

*where*

$$\mathcal{L}_1(f) = \sum_{i \in \mathcal{Y}_l} \underset{\substack{\bar{x}_l \sim \mathcal{P}_{l_i}, \bar{x}_l' \sim \mathcal{P}_{l_i}, \\ x \sim \mathcal{T}(\cdot|\bar{x}_l), x^+ \sim \mathcal{T}(\cdot|\bar{x}_l')}}{\mathbb{E}} \left[ f(x)^\top f\left(x^+\right) \right], \mathcal{L}_2(f) = \underset{\substack{\bar{x}_u \sim \mathcal{P}, \\ x \sim \mathcal{T}(\cdot|\bar{x}_u), x^+ \sim \mathcal{T}(\cdot|\bar{x}_u)}}{\mathbb{E}} \left[ f(x)^\top f\left(x^+\right) \right],$$

$$\mathcal{L}_3(f) = \sum_{i,j \in \mathcal{Y}_l} \underset{\substack{\bar{x}_l \sim \mathcal{P}_{l_i}, \bar{x}_l' \sim \mathcal{P}_{l_j}, \\ x \sim \mathcal{T}(\cdot|\bar{x}_l), x^- \sim \mathcal{T}(\cdot|\bar{x}_l')}}{\mathbb{E}} \left[ \left( f(x)^\top f\left(x^-\right) \right)^2 \right],$$

$$\mathcal{L}_4(f) = \sum_{i \in \mathcal{Y}_l} \underset{\substack{\bar{x}_l \sim \mathcal{P}_{l_i}, \bar{x}_u \sim \mathcal{P}, \\ x \sim \mathcal{T}(\cdot|\bar{x}_l), x^- \sim \mathcal{T}(\cdot|\bar{x}_u)}}{\mathbb{E}} \left[ \left( f(x)^\top f\left(x^-\right) \right)^2 \right], \mathcal{L}_5(f) = \underset{\substack{\bar{x}_u \sim \mathcal{P}, \bar{x}_u' \sim \mathcal{P}, \\ x \sim \mathcal{T}(\cdot|\bar{x}_u), x^- \sim \mathcal{T}(\cdot|\bar{x}_u')}}{\mathbb{E}} \left[ \left( f(x)^\top f\left(x^-\right) \right)^2 \right].$$

---

[2]Theorem 3.1 is primarily adapted from Theorem 4.1 in [64]. However, there is a distinction in the data setting, as Sun et al. [64] do not consider known class samples within the unlabeled dataset.

*Proof.* (*sketch*) We can expand $\mathcal{L}_{\mathrm{mf}}(F, A)$ and obtain

$$\mathcal{L}_{\mathrm{mf}}(F, A) = \sum_{x,x'\in\mathcal{X}}\left(\frac{w_{xx'}}{\sqrt{w_x w_{x'}}} - \mathbf{f}_x^\top \mathbf{f}_{x'}\right)^2 = const + \sum_{x,x'\in\mathcal{X}}\left(-2w_{xx'}f(x)^\top f(x') + w_x w_{x'}\left(f(x)^\top f(x')\right)^2\right)$$

The form of $\mathcal{L}_{\mathrm{SORL}}(f)$ is derived from plugging $w_{xx'}$ (defined in Eq. (1)) and $w_x$. Full proof is in Appendix A. $\qquad\square$

**Interpretation of $\mathcal{L}_{\mathbf{SORL}}(f)$.** At a high level, $\mathcal{L}_1$ and $\mathcal{L}_2$ push the embeddings of **positive pairs** to be closer while $\mathcal{L}_3$, $\mathcal{L}_4$ and $\mathcal{L}_5$ pull away the embeddings of **negative pairs**. In particular, $\mathcal{L}_1$ samples two random augmentation views of two images from labeled data with the **same** class label, and $\mathcal{L}_2$ samples two views from the same image in $\mathcal{X}$. For negative pairs, $\mathcal{L}_3$ uses two augmentation views from two samples in $\mathcal{X}_l$ with **any** class label. $\mathcal{L}_4$ uses two views of one sample in $\mathcal{X}_l$ and another one in $\mathcal{X}$. $\mathcal{L}_5$ uses two views from two random samples in $\mathcal{X}$. This training objective, though bearing similarities to NSCL [64], operates within a distinct problem domain. Accordingly, we derive novel theoretical analysis uniquely tailored to our problem setting, which we present next.

## 4 Theoretical Analysis

So far we have presented a spectral approach for open-world semi-supervised learning based on graph factorization. Under this framework, we now formally analyze: ***how does the labeling information shape the representations for known and novel classes?***

### 4.1 An Illustrative Example

We consider a toy example that helps illustrate the core idea of our theoretical findings. Specifically, the example aims to distinguish 3D objects with different shapes, as shown in Figure 2. These images are generated by a 3D rendering software [31] with user-defined properties including colors, shape, size, position, etc. We are interested in contrasting the representations (in the form of singular vectors), when the label information is either incorporated in training or not.

**Data design.** Suppose the training samples come from three types, $\mathcal{X}_\boxminus, \mathcal{X}_\bigcirc, \mathcal{X}_\ominus$. Let $\mathcal{X}_\boxminus$ be the sample space with **known** class, and $\mathcal{X}_\bigcirc, \mathcal{X}_\ominus$ be the sample space with **novel** classes. Further, the two novel classes are constructed to have different relationships with the known class. Specifically, $\mathcal{X}_\bigcirc$ shares some similarity with $\mathcal{X}_\boxminus$ in color (red and blue); whereas another novel class $\mathcal{X}_\ominus$ has no obvious similarity with the known class. Without any labeling information, it can be difficult to distinguish $\mathcal{X}_\bigcirc$ from $\mathcal{X}_\boxminus$ since samples share common colors. We aim to verify the hypothesis that: *adding labeling information to $\mathcal{X}_\boxminus$ (i.e., connecting ◻ and ▱) has a larger (beneficial) impact to cluster $\mathcal{X}_\bigcirc$ than $\mathcal{X}_\ominus$.*

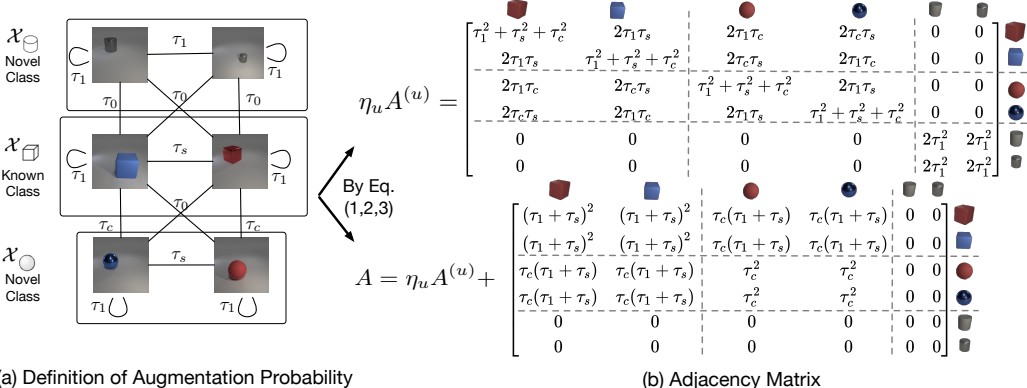

(a) Definition of Augmentation Probability
(b) Adjacency Matrix

Figure 2: An illustrative example for theoretical analysis. We consider a 6-node graph with one known class (cube) and two novel classes (sphere, cylinder). (a) The augmentation probabilities between nodes are defined by their color and shape in Eq. (7). (b) The adjacency matrix can then be calculated by Equations in Sec. 3.1 where we let $\tau_0 = 0, \eta_u = 6, \eta_l = 4$. The calculation details are in Appendix B. The magnitude order follows $\tau_1 \gg \tau_c > \tau_s > 0$.

**Augmentation graph.** Based on the data design, we formally define the augmentation graph, which encodes the probability of augmenting a source image $\bar{x}$ to the augmented view $x$:

$$
\mathcal{T}\left(x \mid \bar{x}\right) = \begin{cases} \tau_1 & \text{if } \mathrm{color}(x) = \mathrm{color}(\bar{x}), \mathrm{shape}(x) = \mathrm{shape}(\bar{x}); \\ \tau_c & \text{if } \mathrm{color}(x) = \mathrm{color}(\bar{x}), \mathrm{shape}(x) \neq \mathrm{shape}(\bar{x}); \\ \tau_s & \text{if } \mathrm{color}(x) \neq \mathrm{color}(\bar{x}), \mathrm{shape}(x) = \mathrm{shape}(\bar{x}); \\ \tau_0 & \text{if } \mathrm{color}(x) \neq \mathrm{color}(\bar{x}), \mathrm{shape}(x) \neq \mathrm{shape}(\bar{x}). \end{cases}
\tag{7}
$$

With Eq. (7) and the definition of the adjacency matrix in Section 3.1, we can derive the analytic form of $A^{(u)}$ and $A$, as shown in Figure 2(b). We refer readers to Appendix B for the detailed derivation. The two matrices allow us to contrast the connectivity changes in the graph, before and after the labeling information is added.**Insights.** We are primarily interested in analyzing the difference of the representation space derived from $A^{(u)}$ and $A$. We visualize the top-3 eigenvectors[3] of the normalized adjacency matrix $\tilde{A}^{(u)}$ and $\tilde{A}$ in Figure 3(a), where the results are based on the magnitude order $\tau_1 \gg \tau_c > \tau_s > 0$. Our key takeaway is: *adding labeling information to known class $\mathcal{X}_{\square}$ helps better distinguish the known class itself and the novel class $\mathcal{X}_{\bigcirc}$, which has a stronger connection/similarity with $\mathcal{X}_{\square}$.*

**Qualitative analysis.** Our theoretical insight can also be verified empirically, by learning representations on over 10,000 samples using the loss defined in Section 3.2. Due to the space limitation, we include experimental details in Appendix E.1. In Figure 3(b), we visualize the learned features through UMAP [43]. Indeed, we observe that samples become more concentrated around different shape classes after adding labeling information to the cube class.

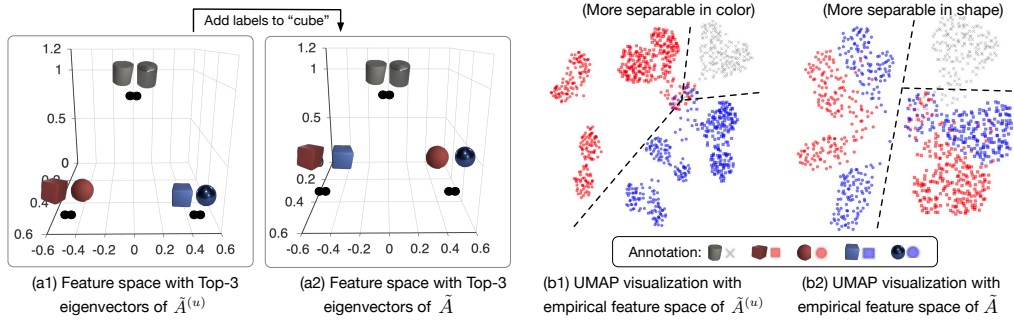

(a1) Feature space with Top-3 eigenvectors of $\tilde{A}^{(u)}$    (a2) Feature space with Top-3 eigenvectors of $\tilde{A}$    (b1) UMAP visualization with empirical feature space of $\tilde{A}^{(u)}$    (b2) UMAP visualization with empirical feature space of $\tilde{A}$

Figure 3: Visualization of representation space for toy example. (a) Theoretically contrasting the feature formed by top-3 eigenvectors of $\tilde{A}^{(u)}$ and $\tilde{A}$ respectively. (b) UMAP visualization of the features learned without (left) and with labeled information (right). Details are in Appendix B (eigenvector calculation) and Appendix E.1 (visualization setting).

## 4.2 Main Theory

The toy example offers an important insight that the added labeled information is more helpful for the class with a stronger connection to the known class. In this section, we formalize this insight by extending the toy example to a more general setting. As a roadmap, we derive the result through three steps: **(1)** derive the closed-form solution of the learned representations; **(2)** define the clustering performance by the K-means measure; **(3)** contrast the resulting clustering performance before and after adding labels. We start by deriving the representations.

### 4.2.1 Learned Representations in Analytic Form

**Representation without labels.** To obtain the representations, one can train the neural network $f : \mathcal{X} \mapsto \mathbb{R}^k$ using the spectral loss defined in Equation 6. We assume that the optimizer is capable to obtain the representation $Z^{(u)} \in \mathbb{R}^{N \times k}$ that minimizes the loss, where each row vector $\mathbf{z}_i = f(x_i)^\top$. Recall that Theorem 3.1 allows us to derive a closed-form solution for the learned feature space by the spectral decomposition of the adjacency matrix, which is $\tilde{A}^{(u)}$ in the case without labeling information. Specifically, we have $F_k^{(u)} = \sqrt{D^{(u)}} Z^{(u)}$, where $F_k^{(u)} F_k^{(u)\top}$ contains the

---

[3]When $\tau_1 \gg \tau_c > \tau_s > 0$, the top-3 eigenvectors are almost equivalent to the feature embedding.

top-$k$ components of $\tilde{A}^{(u)}$'s SVD decomposition and $D^{(u)}$ is the diagonal matrix defined based on the row sum of $A^{(u)}$. We further define the top-$k$ singular vectors of $\tilde{A}^{(u)}$ as $V_k^{(u)} \in \mathbb{R}^{N \times k}$, so we have $F_k^{(u)} = V_k^{(u)}\sqrt{\Sigma_k^{(u)}}$, where $\Sigma_k^{(u)}$ is a diagonal matrix of the top-$k$ singular values of $\tilde{A}^{(u)}$. By equalizing the two forms of $F_k^{(u)}$, the closed-formed solution of the learned feature space is given by $Z^{(u)} = [D^{(u)}]^{-\frac{1}{2}} V_k^{(u)}\sqrt{\Sigma_k^{(u)}}$.

**Representation perturbation by adding labels.** We now analyze how the representation is "perturbed" as a result of adding label information. We consider $|\mathcal{Y}_l| = 1$[4] to facilitate a better understanding of our key insight. We can rewrite $A$ in Eq. 3 as:

$$A(\delta) \triangleq \eta_u A^{(u)} + \delta \mathfrak{l}^\top,$$

where we replace $\eta_l$ to $\delta$ to be more apparent in representing the perturbation and define $\mathfrak{l} \in \mathbb{R}^N$, $(\mathfrak{l})_x = \mathbb{E}_{\bar{x}_l \sim \mathcal{P}_{l_1}} \mathcal{T}(x|\bar{x}_l)$. Note that $\mathfrak{l}$ can be interpreted as the vector of "*the semantic connection for sample $x$ to the labeled data*". One can easily extend to $r$ classes by letting $\mathfrak{l} \in \mathbb{R}^{N \times r}$.

Here we treat the adjacency matrix as a function of the perturbation. In a similar manner as above, we can derive the normalized adjacency matrix $\tilde{A}(\delta)$ and the feature representation $Z(\delta)$ in closed form. The details are included in Appendix C.4.

### 4.2.2 Evaluation Target

With the learned representations, we can evaluate their quality by the clustering performance. Our theoretical analysis of the clustering performance can well connect to empirical evaluation strategy in the literature [75] using $K$-means clustering accuracy/error. Formally, we define the ground-truth partition of clusters by $\Pi = \{\pi_1, \pi_2, ..., \pi_C\}$, where $\pi_i$ is the set of samples' indices with underlying label $y_i$ and $C$ is the total number of classes (including both known and novel). We further let $\boldsymbol{\mu}_\pi = \mathbb{E}_{i \in \pi} \mathbf{z}_i$ be the center of features in $\pi$, and the average of all feature vectors be $\boldsymbol{\mu}_\Pi = \mathbb{E}_{j \in [N]} \mathbf{z}_j$.

The clustering performance of K-means depends on two measurements: **Intra-class** measure and **Inter-class** measure. Specifically, we let the intra-class measure be the average Euclidean distance from the samples' feature to the corresponding cluster center and we measure the inter-class separation as the distances between cluster centers:

$$\mathcal{M}_{\text{intra-class}}(\Pi, Z) \triangleq \sum_{\pi \in \Pi} \sum_{i \in \pi} \|\mathbf{z}_i - \boldsymbol{\mu}_\pi\|^2, \mathcal{M}_{\text{inter-class}}(\Pi, Z) \triangleq \sum_{\pi \in \Pi} |\pi| \|\boldsymbol{\mu}_\pi - \boldsymbol{\mu}_\Pi\|^2. \tag{8}$$

Strong clustering results translate into low $\mathcal{M}_{\text{intra-class}}$ and high $\mathcal{M}_{\text{inter-class}}$. Thus we define the **K-means measure** as:

$$\mathcal{M}_{kms}(\Pi, Z) \triangleq \mathcal{M}_{\text{intra-class}}(\Pi, Z) / \mathcal{M}_{\text{inter-class}}(\Pi, Z). \tag{9}$$

We also formally show in Theorem 4.1 that the K-means clustering error[5] is asymptotically equivalent to the K-means measure we defined above.

**Theorem 4.1.** *(**Relationship between the K-means measure and K-means error**.) We define the $\xi_{\pi \to \pi'}$ as the index set of samples that is from class division $\pi$ however is closer to $\boldsymbol{\mu}_{\pi'}$ than $\boldsymbol{\mu}_\pi$. In other word, $\xi_{\pi \to \pi'} = \{i : i \in \pi, \|\mathbf{z}_i - \boldsymbol{\mu}_\pi\|_2 \geq \|\mathbf{z}_i - \boldsymbol{\mu}_{\pi'}\|_2\}$. Assuming $|\xi_{\pi \to \pi'}| > 0$, we define below the clustering error ratio from $\pi$ to $\pi'$ as $\mathcal{E}_{\pi \to \pi'}$ and the overall cluster error ratio $\mathcal{E}_{\Pi, Z}$ as the Harmonic Mean of $\mathcal{E}_{\pi \to \pi'}$ among all class pairs:*

$$\mathcal{E}_{\Pi, Z} = C(C-1) / \left( \sum_{\substack{\pi \neq \pi' \\ \pi, \pi' \in \Pi}} \frac{1}{\mathcal{E}_{\pi \to \pi'}} \right), where \ \mathcal{E}_{\pi \to \pi'} = \frac{|\xi_{\pi \to \pi'}|}{|\pi'| + |\pi|}.$$

*The K-means measure $\mathcal{M}_{kms}(\Pi, Z)$ has the same order of the Harmonic Mean of the cluster error ratio between all cluster pairs with proof in Appendix C.3.*

$$\mathcal{E}_{\Pi, Z} = O(\mathcal{M}_{kms}(\Pi, Z)).$$

---

[4]To understand the perturbation by adding labels from more than one class, one can take the summation of the perturbation by each class.

[5]It is theoretically inconvenient to directly analyze the clustering error since it is a non-differentiable target.

The K-means measure $\mathcal{M}_{kms}(\Pi, Z)$ have a nice matrix form as shown in Appendix C.2 which facilitates theoretical analysis. Our analysis revolves around contrasting the resulting clustering performance before and after adding labels as we will shown next.

### 4.2.3 Perturbation in Clustering Performance

With the evaluation target defined above, our main analysis will revolve around analyzing *"how the extra label information help reduces $\mathcal{M}_{kms}(\Pi, Z)$"*. Formally, we investigate the following error difference, as a result of added label information:

$$\Delta_{kms}(\delta) = \mathcal{M}_{kms}(\Pi, Z) - \mathcal{M}_{kms}(\Pi, Z(\delta)),$$

where the closed-form solution is given by the following theorem. Positive $\Delta_{kms}(\delta)$ means improved clustering, as a result of adding labeling information.

---

**Theorem 4.2.** *(Main result.) Denote $V_{\varnothing}^{(u)} \in \mathbb{R}^{N \times (N-k)}$ as the null space of $V_k^{(u)}$ and $\tilde{A}_k^{(u)} = V_k^{(u)} \Sigma_k^{(u)} V_k^{(u)\top}$ as the rank-k approximation for $\tilde{A}^{(u)}$. Given $\delta, \eta_1 > 0$ and let $\mathcal{G}_k$ as the spectral gap between k-th and k + 1-th singular values of $\tilde{A}^{(u)}$, we have:*

$$\Delta_{kms}(\delta) = \delta \eta_1 \operatorname{Tr}\left(\Upsilon\left(V_k^{(u)} V_k^{(u)\top} \mathfrak{l}^\top (I + V_{\varnothing}^{(u)} V_{\varnothing}^{(u)\top}) - 2\tilde{A}_k^{(u)} diag(\mathfrak{l})\right)\right) + O\left(\frac{1}{\mathcal{G}_k} + \delta^2\right),$$

*where $diag(\cdot)$ converts the vector to the corresponding diagonal matrix and $\Upsilon \in \mathbb{R}^{N \times N}$ is a matrix encoding the **ground-truth clustering structure** in the way that $\Upsilon_{xx'} > 0$ if $x$ and $x'$ has the same label and $\Upsilon_{xx'} < 0$ otherwise. The concrete form and the proof are in Appendix C.4.*

---

Theorem 4.2 is more general but less intuitive to understand. To gain a better insight, we introduce Theorem 4.3 which provides more direct implications. We provide the justification of the assumptions and the formal proof in Appendix C.5.

---

**Theorem 4.3.** *(Intuitive result.) Assuming the spectral gap $\mathcal{G}_k$ is sufficiently large and $\mathfrak{l}$ lies in the linear span of $V_k^{(u)}$. We also assume $\forall \pi_c \in \Pi, \forall i \in \pi_c, \mathfrak{l}_{(i)} =: \mathfrak{l}_{\pi_c}$ which represents the connection between class c to the labeled data. Given $\delta, \eta_1, \eta_2 > 0$, we have:*

$$\Delta_{kms}(\delta) \geq \delta \eta_1 \eta_2 \sum_{\pi_c \in \Pi} |\pi_c| \mathfrak{l}_{\pi_c} \Delta_{\pi_c}(\delta),$$

*where*

                    *Connection from class c to the labeled data.*

$$\Delta_{\pi_c}(\delta) = \underbrace{\left(\mathfrak{l}_{\pi_c} - \frac{1}{N}\right)} - 2\left(1 - \frac{|\pi_c|}{N}\right)\left(\underbrace{\mathbb{E}_{i \in \pi_c}\mathbb{E}_{j \in \pi_c} \mathbf{z}_i^\top \mathbf{z}_j} - \underbrace{\mathbb{E}_{i \in \pi_c}\mathbb{E}_{j \notin \pi_c} \mathbf{z}_i^\top \mathbf{z}_j}\right).$$

                          *Intra-class similarity ↑*                  *↑Inter-class similarity*

---

**Implications.** In Theorem 4.3, we define the **class-wise perturbation** of the K-means measure as $\Delta_{\pi_c}(\delta)$. This way, we can interpret the effect of adding labels for a specific class $c$. If we desire $\Delta_{\pi_c}(\delta)$ to be large, the sufficient condition is that

    *connection of class c to the labeled data > intra-class similarity - inter-class similarity.*

We use examples in Figure 1 to epitomize the core idea. Specifically, our unlabeled samples consist of three underlying classes: traffic lights (known), apples (novel), and flowers (novel). **(a)** For unlabeled traffic lights from *known classes* which are strongly connected to the labeled data, adding labels to traffic lights can largely improve the clustering performance; **(b)** For *novel classes* like apples, it may also help when they have a strong connection to the traffic light, and their intra-class similarity is not as strong (due to different colors); **(c)** However, labeled data may offer little improvement in clustering the flower class, due to the minimal connection to the labeled data and that flowers' self-clusterability is already strong.

# 5 Empirical Validation of Theory

Beyond theoretical insights, we show empirically that SORL is effective on standard benchmark image classification datasets CIFAR-10/100 [35]. Following the seminal work ORCA [7], classes are divided into 50% known and 50% novel classes. We then use 50% of samples from the known classes as the labeled dataset, and the rest as the unlabeled set. We follow the evaluation strategy in [7] and report the following metrics: (1) classification accuracy on known classes, (2) clustering accuracy on the novel data, and (3) overall accuracy on all classes. More experiment details are in Appendix E.2.

Table 1: Main Results. Mean and std are estimated on five different runs. Baseline numbers are from [7, 63].

| Method | CIFAR-10 | | | CIFAR-100 | | |
|---|---|---|---|---|---|---|
| | All | Novel | Known | All | Novel | Known |
| **FixMatch** [37] | 49.5 | 50.4 | 71.5 | 20.3 | 23.5 | 39.6 |
| **DS$^3$L** [21] | 40.2 | 45.3 | 77.6 | 24.0 | 23.7 | 55.1 |
| **CGDL** [62] | 39.7 | 44.6 | 72.3 | 23.6 | 22.5 | 49.3 |
| **DTC** [22] | 38.3 | 39.5 | 53.9 | 18.3 | 22.9 | 31.3 |
| **RankStats** [82] | 82.9 | 81.0 | 86.6 | 23.1 | 28.4 | 36.4 |
| **SimCLR** [11] | 51.7 | 63.4 | 58.3 | 22.3 | 21.2 | 28.6 |
| **ORCA** [7] | $88.3^{\pm 0.3}$ | $87.5^{\pm 0.2}$ | $89.9^{\pm 0.4}$ | $47.2^{\pm 0.7}$ | $41.0^{\pm 1.0}$ | $66.7^{\pm 0.2}$ |
| **GCD** [69] | $87.5^{\pm 0.5}$ | $86.7^{\pm 0.4}$ | $90.1^{\pm 0.3}$ | $46.8^{\pm 0.5}$ | $43.4^{\pm 0.7}$ | $69.7^{\pm 0.4}$ |
| **OpenCon** [63] | $90.4^{\pm 0.6}$ | $91.1^{\pm 0.1}$ | $89.3^{\pm 0.2}$ | $52.7^{\pm 0.6}$ | $47.8^{\pm 0.6}$ | $69.1^{\pm 0.3}$ |
| **SORL (Ours)** | $\mathbf{93.5}^{\pm 1.0}$ | $\mathbf{92.5}^{\pm 0.1}$ | $\mathbf{94.0}^{\pm 0.2}$ | $\mathbf{56.1}^{\pm 0.3}$ | $\mathbf{52.0}^{\pm 0.2}$ | $68.2^{\pm 0.1}$ |

**SORL achieves competitive performance.** Our proposed loss SORL is amenable to the theoretical understanding, which is our primary goal of this work. Beyond theory, we show that SORL is equally desirable in empirical performance. In particular, SORL displays competitive performance compared to existing methods, as evidenced in Table 1. Our comparison covers an extensive collection of very recent algorithms developed for this problem, including ORCA [7], GCD [69], and OpenCon [63]. We also compare methods in related problem domains: (1) Semi-Supervised Learning [21, 37, 62], (2) Novel Class Discovery [22, 82], (3) common representation learning method SimCLR [11]. In particular, on CIFAR-100, we improve upon the best baseline OpenCon by **3.4%** in terms of overall accuracy. Our result further validates that putting analysis on SORL is appealing for both theoretical and empirical reasons.

# 6 Broader Impact

From a theoretical perspective, our graph-theoretic framework can facilitate and deepen the understanding of other representation learning methods that commonly involve the notion of positive/negative pairs. In Appendix D, *we exemplify how our framework can be potentially generalized to other common contrastive loss functions* [11, 34, 68], and baseline methods that are tailored for the open-world semi-supervised learning problem (e.g., GCD [69], OpenCon [63]). Hence, we believe our theoretical framework has a broader utility and significance.

From a practical perspective, our work can directly impact and benefit many real-world applications, where unlabeled data are produced at an incredible rate today. Major companies exhibit a strong need for making their machine learning systems and services amendable for the open-world setting but lack fundamental and systematic knowledge. Hence, our research advances the understanding of open-world machine learning and helps the industry improve ML systems by discovering insights and structures from unlabeled data.

# 7 Related Work

**Semi-supervised learning.** Semi-supervised learning (SSL) is a classic problem in machine learning. SSL typically assumes the same class space between labeled and unlabeled data, and hence remains closed-world. A rich line of empirical works [9, 13, 21, 29, 37, 38, 39, 42, 48, 50, 53, 54, 74, 76, 78] and theoretical efforts [3, 46, 47, 51, 60, 61, 73] have been made to address this problem. An important class of SSL methods is to represent data as graphs and predict labels by aggregating proximal nodes' labels [1, 18, 30, 71, 80, 84, 85]. Different from classic SSL, we allow its semantic space to cover both known and novel classes. Accordingly, we contribute a graph-theoretic framework tailored to the open-world setting, and reveal new insights on how the labeled data can benefit the clustering performance on both known and novel classes.

**Open-world semi-supervised learning**. The learning setting that considers both labeled and un-labeled data with a mixture of known and novel classes is first proposed in [7] and inspires a proliferation of follow-up works [49, 52, 63, 69, 81] advancing empirical success. Most works put emphasis on learning high-quality embeddings [49, 63, 69, 81]. In particular, Sun and Li [63] employ contrastive learning with both supervised and self-supervised signals, which aligns with our theoretical setup in Sec. 3.1. Different from prior works, our paper focuses on *advancing theoretical understanding*. To the best of our knowledge, we are the first to theoretically investigate the problem from a graph-theoretic perspective and provide a rigorous error bound.

**Spectral graph theory.** Spectral graph theory is a classic research problem [10, 14, 33, 40, 44, 70], which aims to partition the graph by studying the eigenspace of the adjacency matrix. The spectral graph theory is also widely applied in machine learning [1, 6, 45, 56, 58, 64, 86]. Recently, HaoChen et al. [23] derive a spectral contrastive loss from the factorization of the graph's adjacency matrix which facilitates theoretical study in unsupervised domain adaptation [24, 57]. In these works, the graph's formulation is exclusively based on unlabeled data. Sun et al. [64] later expanded this spectral contrastive loss approach to cater to learning environments that encompass both labeled data from known classes and unlabeled data from novel ones. In this paper, our adaptation of the loss function from [64] is tailored to address the open-world semi-supervised learning challenge, considering known class samples within unlabeled data.

**Theory for self-supervised learning.** A proliferation of works in self-supervised representation learning demonstrates the empirical success [5, 8, 11, 12, 23, 26, 68, 77] with the theoretical foundation by providing provable guarantees on the representations learned by contrastive learning for linear probing [4, 41, 55, 59, 66, 67]. From the graphic view, HaoChen et al. [23, 24], Shen et al. [57] model the pairwise relation by the augmentation probability and provided error analysis of the downstream tasks. The existing body of work has mostly focused on *unsupervised learning*. In this paper, we systematically investigate how the label information can change the representation manifold and affect the downstream clustering performance on both known and novel classes.

## 8 Conclusion

In this paper, we present a graph-theoretic framework for open-world semi-supervised learning. The framework facilitates the understanding of how representations change as a result of adding labeling information to the graph. Specifically, we learn representation through Spectral Open-world Representation Learning (SORL). Minimizing this objective is equivalent to factorizing the graph's adjacency matrix, which allows us to analyze the clustering error difference between having vs. excluding labeled data. Our main results suggest that the clustering error can be significantly reduced if the connectivity to the labeled data is stronger than their self-clusterability. Our framework is also empirically appealing to use since it achieves competitive performance on par with existing baselines. Nevertheless, we acknowledge two limitations to practical application within our theoretical construct:

- The augmentation graph serves as a potent theoretical tool for elucidating the success of modern representation learning methods. However, it is challenging to ensure that current augmentation strategies, such as cropping, color jittering, can transform two dissimilar images into identical ones.

- The utilization of Theorems 4.1 and 4.2 necessitates an explicit knowledge of the adjacency matrix of the augmentation graph, a requirement that can be intractable in practice.

In light of these limitations, we encourage further research to enhance the practicality of these theoretical findings. We also hope our framework and insights can inspire the broader representation learning community to understand the role of labeling prior.

## Acknowledgement

Research is supported by the AFOSR Young Investigator Program under award number FA9550-23-1-0184, National Science Foundation (NSF) Award No. IIS-2237037 & IIS-2331669, Office of Naval Research under grant number N00014-23-1-2643, and faculty research awards/gifts from Google and Meta. Any opinions, findings, conclusions, or recommendations expressed in this material are those of the authors and do not necessarily reflect the views, policies, or endorsements either expressed or implied, of the sponsors. The authors would also like to thank Tengyu Ma, Xuefeng Du, and Yifei Ming for their helpful suggestions and feedback.

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

# Appendix

## A Technical Details of Spectral Open-world Representation Learning

**Theorem A.1.** *(Recap of Theorem 3.1) We define* $\mathbf{f}_x = \sqrt{w_x} f(x)$ *for some function* $f$. *Recall* $\eta_u, \eta_l$ *are two hyper-parameters defined in Eq.* (1). *Then minimizing the loss function* $\mathcal{L}_{\mathrm{mf}}(F, A)$ *is equivalent to minimizing the following loss function for* $f$, *which we term **Spectral Open-world Representation Learning (SORL)**:*

$$\mathcal{L}_{SORL}(f) \triangleq -2\eta_l \mathcal{L}_1(f) - 2\eta_u \mathcal{L}_2(f) \\ + \eta_l^2 \mathcal{L}_3(f) + 2\eta_l \eta_u \mathcal{L}_4(f) + \eta_u^2 \mathcal{L}_5(f), \tag{10}$$

*where*

$$\mathcal{L}_1(f) = \sum_{i \in \mathcal{Y}_l} \mathbb{E}_{\substack{\bar{x}_l \sim \mathcal{P}_{l_i}, \bar{x}'_l \sim \mathcal{P}_{l_i}, \\ x \sim \mathcal{T}(\cdot|\bar{x}_l), x^+ \sim \mathcal{T}(\cdot|\bar{x}'_l)}} \left[ f(x)^\top f\left(x^+\right) \right],$$

$$\mathcal{L}_2(f) = \mathbb{E}_{\substack{\bar{x}_u \sim \mathcal{P}, \\ x \sim \mathcal{T}(\cdot|\bar{x}_u), x^+ \sim \mathcal{T}(\cdot|\bar{x}_u)}} \left[ f(x)^\top f\left(x^+\right) \right],$$

$$\mathcal{L}_3(f) = \sum_{i,j \in \mathcal{Y}_l} \mathbb{E}_{\substack{\bar{x}_l \sim \mathcal{P}_{l_i}, \bar{x}'_l \sim \mathcal{P}_{l_j}, \\ x \sim \mathcal{T}(\cdot|\bar{x}_l), x^- \sim \mathcal{T}(\cdot|\bar{x}'_l)}} \left[ \left( f(x)^\top f\left(x^-\right) \right)^2 \right],$$

$$\mathcal{L}_4(f) = \sum_{i \in \mathcal{Y}_l} \mathbb{E}_{\substack{\bar{x}_l \sim \mathcal{P}_{l_i}, \bar{x}_u \sim \mathcal{P}, \\ x \sim \mathcal{T}(\cdot|\bar{x}_l), x^- \sim \mathcal{T}(\cdot|\bar{x}_u)}} \left[ \left( f(x)^\top f\left(x^-\right) \right)^2 \right],$$

$$\mathcal{L}_5(f) = \mathbb{E}_{\substack{\bar{x}_u \sim \mathcal{P}, \bar{x}'_u \sim \mathcal{P}, \\ x \sim \mathcal{T}(\cdot|\bar{x}_u), x^- \sim \mathcal{T}(\cdot|\bar{x}'_u)}} \left[ \left( f(x)^\top f\left(x^-\right) \right)^2 \right].$$

*Proof.* We can expand $\mathcal{L}_{\mathrm{mf}}(F, A)$ and obtain

$$\mathcal{L}_{\mathrm{mf}}(F, A) = \sum_{x,x' \in \mathcal{X}} \left( \frac{w_{xx'}}{\sqrt{w_x w_{x'}}} - \mathbf{f}_x^\top \mathbf{f}_{x'} \right)^2$$

$$= \mathrm{const} + \sum_{x,x' \in \mathcal{X}} \left( -2w_{xx'} f(x)^\top f\left(x'\right) + w_x w_{x'} \left( f(x)^\top f\left(x'\right) \right)^2 \right),$$

where $\mathbf{f}_x = \sqrt{w_x} f(x)$ is a re-scaled version of $f(x)$. At a high level, we follow the proof in [23], while the specific form of loss varies with the different definitions of positive/negative pairs. The form of $\mathcal{L}_{\mathrm{SORL}}(f)$ is derived from plugging $w_{xx'}$ and $w_x$.

Recall that $w_{xx'}$ is defined by

$$w_{xx'} = \eta_l \sum_{i \in \mathcal{Y}_l} \mathbb{E}_{\bar{x}_l \sim \mathcal{P}_{l_i}} \mathbb{E}_{\bar{x}'_l \sim \mathcal{P}_{l_i}} \mathcal{T}(x|\bar{x}_l) \mathcal{T}\left(x'|\bar{x}'_l\right) + \eta_u \mathbb{E}_{\bar{x}_u \sim \mathcal{P}} \mathcal{T}(x|\bar{x}_u) \mathcal{T}\left(x'|\bar{x}_u\right),$$

and $w_x$ is given by

$$w_x = \sum_{x'} w_{xx'}$$

$$= \eta_l \sum_{i \in \mathcal{Y}_l} \mathbb{E}_{\bar{x}_l \sim \mathcal{P}_{l_i}} \mathbb{E}_{\bar{x}'_l \sim \mathcal{P}_{l_i}} \mathcal{T}(x|\bar{x}_l) \sum_{x'} \mathcal{T}\left(x'|\bar{x}'_l\right) + \eta_u \mathbb{E}_{\bar{x}_u \sim \mathcal{P}} \mathcal{T}(x|\bar{x}_u) \sum_{x'} \mathcal{T}\left(x'|\bar{x}_u\right)$$

$$= \eta_l \sum_{i \in \mathcal{Y}_l} \mathbb{E}_{\bar{x}_l \sim \mathcal{P}_{l_i}} \mathcal{T}(x|\bar{x}_l) + \eta_u \mathbb{E}_{\bar{x}_u \sim \mathcal{P}} \mathcal{T}(x|\bar{x}_u).$$

Plugging in $w_{xx'}$ we have,

$$- 2 \sum_{x,x' \in \mathcal{X}} w_{xx'} f(x)^\top f(x')$$

$$= -2 \sum_{x,x^+ \in \mathcal{X}} w_{xx^+} f(x)^\top f\left(x^+\right)$$

$$= -2\eta_l \sum_{i \in \mathcal{Y}_l} \mathbb{E}_{\bar{x}_l \sim \mathcal{P}_{l_i}} \mathbb{E}_{\bar{x}_l' \sim \mathcal{P}_{l_i}} \sum_{x,x' \in \mathcal{X}} \mathcal{T}(x|\bar{x}_l) \mathcal{T}(x'|\bar{x}_l') f(x)^\top f(x')$$
$$- 2\eta_u \mathbb{E}_{\bar{x}_u \sim \mathcal{P}} \sum_{x,x'} \mathcal{T}(x|\bar{x}_u) \mathcal{T}(x'|\bar{x}_u) f(x)^\top f(x')$$

$$= -2\eta_l \sum_{i \in \mathcal{Y}_l} \mathbb{E}_{\substack{\bar{x}_l \sim \mathcal{P}_{l_i}, \bar{x}_l' \sim \mathcal{P}_{l_i}, \\ x \sim \mathcal{T}(\cdot|\bar{x}_l), x^+ \sim \mathcal{T}(\cdot|\bar{x}_l')}} \left[ f(x)^\top f\left(x^+\right) \right]$$
$$- 2\eta_u \mathbb{E}_{\substack{\bar{x}_u \sim \mathcal{P}, \\ x \sim \mathcal{T}(\cdot|\bar{x}_u), x^+ \sim \mathcal{T}(\cdot|\bar{x}_u)}} \left[ f(x)^\top f\left(x^+\right) \right]$$

$$= -2\eta_l \mathcal{L}_1(f) - 2\eta_u \mathcal{L}_2(f).$$

Plugging $w_x$ and $w_{x'}$ we have,

$$\sum_{x,x' \in \mathcal{X}} w_x w_{x'} \left( f(x)^\top f\left(x'\right) \right)^2$$

$$= \sum_{x,x^- \in \mathcal{X}} w_x w_{x^-} \left( f(x)^\top f\left(x^-\right) \right)^2$$

$$= \sum_{x,x' \in \mathcal{X}} \left( \eta_l \sum_{i \in \mathcal{Y}_l} \mathbb{E}_{\bar{x}_l \sim \mathcal{P}_{l_i}} \mathcal{T}(x|\bar{x}_l) + \eta_u \mathbb{E}_{\bar{x}_u \sim \mathcal{P}} \mathcal{T}(x|\bar{x}_u) \right)$$
$$\cdot \left( \eta_l \sum_{j \in \mathcal{Y}_l} \mathbb{E}_{\bar{x}_l' \sim \mathcal{P}_{l_j}} \mathcal{T}(x^-|\bar{x}_l') + \eta_u \mathbb{E}_{\bar{x}_u' \sim \mathcal{P}} \mathcal{T}(x^-|\bar{x}_u') \right) \left( f(x)^\top f\left(x^-\right) \right)^2$$

$$= \eta_l^2 \sum_{x,x^- \in \mathcal{X}} \sum_{i \in \mathcal{Y}_l} \mathbb{E}_{\bar{x}_l \sim \mathcal{P}_{l_i}} \mathcal{T}(x|\bar{x}_l) \sum_{j \in \mathcal{Y}_l} \mathbb{E}_{\bar{x}_l' \sim \mathcal{P}_{l_j}} \mathcal{T}(x^-|\bar{x}_l') \left( f(x)^\top f\left(x^-\right) \right)^2$$
$$+ 2\eta_u \eta_l \sum_{x,x^- \in \mathcal{X}} \sum_{i \in \mathcal{Y}_l} \mathbb{E}_{\bar{x}_l \sim \mathcal{P}_{l_i}} \mathcal{T}(x|\bar{x}_l) \mathbb{E}_{\bar{x}_u \sim \mathcal{P}} \mathcal{T}(x^-|\bar{x}_u) \left( f(x)^\top f\left(x^-\right) \right)^2$$
$$+ \eta_u^2 \sum_{x,x^- \in \mathcal{X}} \mathbb{E}_{\bar{x}_u \sim \mathcal{P}} \mathcal{T}(x|\bar{x}_u) \mathbb{E}_{\bar{x}_u' \sim \mathcal{P}} \mathcal{T}(x^-|\bar{x}_u') \left( f(x)^\top f\left(x^-\right) \right)^2$$

$$= \eta_l^2 \sum_{i \in \mathcal{Y}_l} \sum_{j \in \mathcal{Y}_l} \mathbb{E}_{\substack{\bar{x}_l \sim \mathcal{P}_{l_i}, \bar{x}_l' \sim \mathcal{P}_{l_j}, \\ x \sim \mathcal{T}(\cdot|\bar{x}_l), x^- \sim \mathcal{T}(\cdot|\bar{x}_l')}} \left[ \left( f(x)^\top f\left(x^-\right) \right)^2 \right]$$
$$+ 2\eta_u \eta_l \sum_{i \in \mathcal{Y}_l} \mathbb{E}_{\substack{\bar{x}_l \sim \mathcal{P}_{l_i}, \bar{x}_u \sim \mathcal{P}, \\ x \sim \mathcal{T}(\cdot|\bar{x}_l), x^- \sim \mathcal{T}(\cdot|\bar{x}_u)}} \left[ \left( f(x)^\top f\left(x^-\right) \right)^2 \right]$$
$$+ \eta_u^2 \mathbb{E}_{\substack{\bar{x}_u \sim \mathcal{P}, \bar{x}_u' \sim \mathcal{P}, \\ x \sim \mathcal{T}(\cdot|\bar{x}_u), x^- \sim \mathcal{T}(\cdot|\bar{x}_u')}} \left[ \left( f(x)^\top f\left(x^-\right) \right)^2 \right]$$

$$= \eta_l^2 \mathcal{L}_3(f) + 2\eta_u \eta_l \mathcal{L}_4(f) + \eta_u^2 \mathcal{L}_5(f).$$

$\square$

# B Technical Details for Toy Example

## B.1 Calculation Details for Figure 2.

We first recap the toy example, which illustrates the core idea of our theoretical findings. Specifically, the example aims to distinguish 3D objects with different shapes, as shown in Figure 2. These images are generated by a 3D rendering software [31] with user-defined properties including colors, shape, size, position, etc.

**Data design.** Suppose the training samples come from three types, $\mathcal{X}_{\square}, \mathcal{X}_{\bigcirc}, \mathcal{X}_{\ominus}$. Let $\mathcal{X}_{\square}$ be the sample space with **known** class, and $\mathcal{X}_{\bigcirc}, \mathcal{X}_{\ominus}$ be the sample space with **novel** classes. Further, the two novel classes are constructed to have different relationships with the known class. Specifically, we construct the toy dataset with 6 elements as shown in Figure 4(a).

**Augmentation graph.** Based on the data design, we formally define the augmentation graph, which encodes the probability of augmenting a source image $\bar{x}$ to the augmented view $x$:

$$
\mathcal{T}\left(x \mid \bar{x}\right) = \begin{cases} \tau_1 & \text{if color}(x) = \text{color}(\bar{x}), \text{shape}(x) = \text{shape}(\bar{x}); \\ \tau_c & \text{if color}(x) = \text{color}(\bar{x}), \text{shape}(x) \neq \text{shape}(\bar{x}); \\ \tau_s & \text{if color}(x) \neq \text{color}(\bar{x}), \text{shape}(x) = \text{shape}(\bar{x}); \\ \tau_0 & \text{if color}(x) \neq \text{color}(\bar{x}), \text{shape}(x) \neq \text{shape}(\bar{x}). \end{cases} \tag{11}
$$

According to the definition above, the corresponding augmentation matrix $T$ with each element formed by $\mathcal{T}(\cdot \mid \cdot)$ is given in Figure 4(b). We proceed by showing the details to derive $A^{(u)}$ and $A$ using $T$.

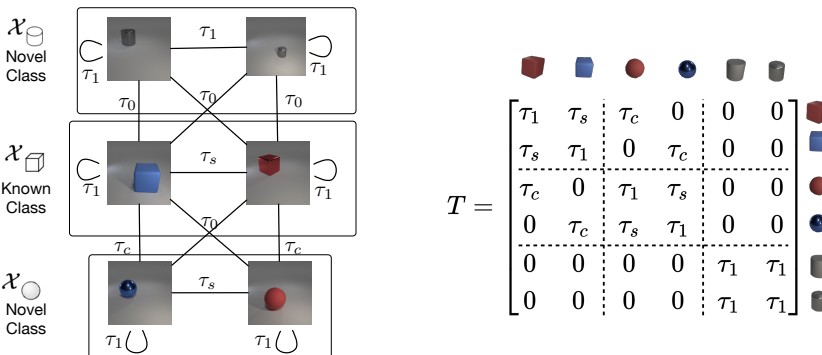

(a) Definition of Augmentation Probability        (b) Augmentation Graph

Figure 4: An illustrative example for theoretical analysis. We consider a 6-node graph with one known class (cube) and two novel classes (sphere, cylinder). (a) The augmentation probabilities between nodes are defined by their color and shape in Eq. (11). (b) The augmentation matrices $T$ derived by Eq. (11) where we let $\tau_0 = 0$.

**Derivation details for $A^{(u)}$ and $A$.** Recall that each element of $A^{(u)}$ is formed by $w_{xx'}^{(u)} = \mathbb{E}_{\bar{x} \sim \mathcal{P}} \mathcal{T}(x|\bar{x}) \mathcal{T}(x'|\bar{x})$. In this toy example, one can then see that $A^{(u)} = \frac{1}{6} T T^\top$ since augmentation matrix $T$ is defined that each element $T_{x\bar{x}} = \mathcal{T}(x|\bar{x})$. Note that $T$ is explicitly given in Figure 4(b) and then if we let $\eta_u = 6$, we have the close-from:

$$
\eta_u A^{(u)} = T^2 = \begin{bmatrix} \tau_1^2 + \tau_s^2 + \tau_c^2 & 2\tau_1\tau_s & 2\tau_1\tau_c & 2\tau_c\tau_s & 0 & 0 \\ 2\tau_1\tau_s & \tau_1^2 + \tau_s^2 + \tau_c^2 & 2\tau_c\tau_s & 2\tau_1\tau_c & 0 & 0 \\ 2\tau_1\tau_c & 2\tau_c\tau_s & \tau_1^2 + \tau_s^2 + \tau_c^2 & 2\tau_1\tau_s & 0 & 0 \\ 2\tau_c\tau_s & 2\tau_1\tau_c & 2\tau_1\tau_s & \tau_1^2 + \tau_s^2 + \tau_c^2 & 0 & 0 \\ 0 & 0 & 0 & 0 & 2\tau_1^2 & 2\tau_1^2 \\ 0 & 0 & 0 & 0 & 2\tau_1^2 & 2\tau_1^2 \end{bmatrix}.
$$

We then derive the second part $A^{(l)}$ whose element is given by:

$$w_{xx'}^{(l)} \triangleq \sum_{i \in \mathcal{Y}_l} \mathbb{E}_{\bar{x}_l \sim \mathcal{P}_{l_i}} \mathbb{E}_{\bar{x}_{l'} \sim \mathcal{P}_{l_i}} \mathcal{T}(x|\bar{x}_l) \mathcal{T}(x'|\bar{x}_l') .$$

Such a form can be simplified in Section 4 by defining $\mathfrak{l} \in \mathbb{R}^N$, $(\mathfrak{l})_x = \mathbb{E}_{\bar{x}_l \sim \mathcal{P}_{l_1}} \mathcal{T}(x|\bar{x}_l)$ and by letting $|\mathcal{Y}_l| = 1$. In this toy example, the known class only has two elements, so $\mathfrak{l} = \frac{1}{2}(T_{:,1} + T_{:,2})$ (average of $T$'s 1st & 2nd column), we then have:

$$A^{(l)} = \mathfrak{l}\mathfrak{l}^\top = \frac{1}{4} \begin{bmatrix} (\tau_1 + \tau_s)^2 & (\tau_1 + \tau_s)^2 & \tau_c(\tau_1 + \tau_s) & \tau_c(\tau_1 + \tau_s) & 0 & 0 \\ (\tau_1 + \tau_s)^2 & (\tau_1 + \tau_s)^2 & \tau_c(\tau_1 + \tau_s) & \tau_c(\tau_1 + \tau_s) & 0 & 0 \\ \tau_c(\tau_1 + \tau_s) & \tau_c(\tau_1 + \tau_s) & \tau_c^2 & \tau_c^2 & 0 & 0 \\ \tau_c(\tau_1 + \tau_s) & \tau_c(\tau_1 + \tau_s) & \tau_c^2 & \tau_c^2 & 0 & 0 \\ 0 & 0 & 0 & 0 & 0 & 0 \\ 0 & 0 & 0 & 0 & 0 & 0 \end{bmatrix} .$$

Finally, if we let $\eta_l = 4$ and $A = \eta_u A^{(u)} + \eta_l A^{(l)}$, we have the full results in Figure 2.

## B.2 Calculation Details for Figure 3.

In this section, we present the analysis of eigenvectors and their orders for toy examples shown in Figure 2. In Theorem B.1 we present the spectral analysis for the adjacency matrix with additional label information while in Theorem B.2, we show the spectral analysis for the unlabeled case.

**Theorem B.1.** *Let*

$$\eta_u A^{(u)} = \begin{bmatrix} \tau_1^2 + \tau_s^2 + \tau_c^2 & 2\tau_1\tau_s & 2\tau_1\tau_c & 2\tau_c\tau_s & 0 & 0 \\ 2\tau_1\tau_s & \tau_1^2 + \tau_s^2 + \tau_c^2 & 2\tau_c\tau_s & 2\tau_1\tau_c & 0 & 0 \\ 2\tau_1\tau_c & 2\tau_c\tau_s & \tau_1^2 + \tau_s^2 + \tau_c^2 & 2\tau_1\tau_s & 0 & 0 \\ 2\tau_c\tau_s & 2\tau_1\tau_c & 2\tau_1\tau_s & \tau_1^2 + \tau_s^2 + \tau_c^2 & 0 & 0 \\ 0 & 0 & 0 & 0 & 2\tau_1^2 & 2\tau_1^2 \\ 0 & 0 & 0 & 0 & 2\tau_1^2 & 2\tau_1^2 \end{bmatrix} ,$$

$$A = \eta_u A^{(u)} + \begin{bmatrix} (\tau_1 + \tau_s)^2 & (\tau_1 + \tau_s)^2 & \tau_c(\tau_1 + \tau_s) & \tau_c(\tau_1 + \tau_s) & 0 & 0 \\ (\tau_1 + \tau_s)^2 & (\tau_1 + \tau_s)^2 & \tau_c(\tau_1 + \tau_s) & \tau_c(\tau_1 + \tau_s) & 0 & 0 \\ \tau_c(\tau_1 + \tau_s) & \tau_c(\tau_1 + \tau_s) & \tau_c^2 & \tau_c^2 & 0 & 0 \\ \tau_c(\tau_1 + \tau_s) & \tau_c(\tau_1 + \tau_s) & \tau_c^2 & \tau_c^2 & 0 & 0 \\ 0 & 0 & 0 & 0 & 0 & 0 \\ 0 & 0 & 0 & 0 & 0 & 0 \end{bmatrix} ,$$

*and we assume that $1 \gg \frac{\tau_c}{\tau_1} > \frac{\tau_s}{\tau_1} > 0$, $\frac{4}{9}\tau_c \leq \tau_s \leq \tau_c$ and $\tau_1 + \tau_c + \tau_s = 1$.*

*Let $\lambda_1, \lambda_2, \lambda_3$ and $v_1, v_2, v_3$ be the largest three eigenvalues and their corresponding eigenvectors of $D^{-\frac{1}{2}} A D^{-\frac{1}{2}}$, which is the normalized adjacency matrix of $A$. Then the concrete form of $\lambda_1, \lambda_2, \lambda_3$ and $v_1, v_2, v_3$ can be approximately given by:*

$$\hat{\lambda}_1 = 1, \ \hat{\lambda}_2 = 1, \ \hat{\lambda}_3 = 1 - \frac{16}{3}\frac{\tau_c}{\tau_1},$$

$$\hat{v}_1 = [0, 0, 0, 0, 1, 1],$$

$$\hat{v}_2 = [\sqrt{3}, \sqrt{3}, 1, 1, 0, 0],$$

$$\hat{v}_3 = [1, 1, -\sqrt{3}, -\sqrt{3}, 0, 0].$$

*Note that the approximation gap can be tightly bounded. Specifically, for $i \in \{1, 2, 3\}$, we have $|\lambda_i - \hat{\lambda}_i| \leq O((\frac{\tau_c}{\tau_1})^2)$ and $\|\sin(U, \hat{U})[6]\|_F \leq O(\frac{\tau_c}{\tau_1})$, where $U = [v_1, v_2, v_3], \hat{U} = [\hat{v}_1, \hat{v}_2, \hat{v}_3]$.*

---

[6]The sin operation measures the distance of two matrices with orthonormal columns, which is usually used in the subspace distance. See more in https://trungvietvu.github.io/notes/2020/DavisKahan.

*Proof.* By $\tau_1 + \tau_c + \tau_s = 1$ and $1 \gg \frac{\tau_c}{\tau_1} > \frac{\tau_s}{\tau_1} > 0$, we define the following equation which approximates the corresponding terms up to error $O((\frac{\tau_c}{\tau_1})^2)$:

$$A \approx \widehat{A} = \tau_1^2 \begin{bmatrix} 2 + 2\frac{\tau_s}{\tau_1} & 1 + 4\frac{\tau_s}{\tau_1} & 3\frac{\tau_c}{\tau_1} & \frac{\tau_c}{\tau_1} & 0 & 0 \\ 1 + 4\frac{\tau_s}{\tau_1} & 2 + 2\frac{\tau_s}{\tau_1} & \frac{\tau_c}{\tau_1} & 3\frac{\tau_c}{\tau_1} & 0 & 0 \\ 3\frac{\tau_c}{\tau_1} & \frac{\tau_c}{\tau_1} & 1 & 2\frac{\tau_s}{\tau_1} & 0 & 0 \\ \frac{\tau_c}{\tau_1} & 3\frac{\tau_c}{\tau_1} & 2\frac{\tau_s}{\tau_1} & 1 & 0 & 0 \\ 0 & 0 & 0 & 0 & 2 & 2 \\ 0 & 0 & 0 & 0 & 2 & 2 \end{bmatrix}.$$

$$D \approx \widehat{D} = \tau_1^2 diag\left(\left[3\left(1 + 2\frac{\tau_s}{\tau_1} + \frac{4}{3}\frac{\tau_c}{\tau_1}\right), 3\left(1 + 2\frac{\tau_s}{\tau_1} + \frac{4}{3}\frac{\tau_c}{\tau_1}\right), 1 + 2\frac{\tau_s}{\tau_1} + 4\frac{\tau_c}{\tau_1}, 1 + 2\frac{\tau_s}{\tau_1} + 4\frac{\tau_c}{\tau_1}, 4, 4\right]\right).$$

$$D^{-\frac{1}{2}} \approx \widehat{D^{-\frac{1}{2}}} = \frac{1}{\tau_1} diag\left(\left[\sqrt{3}\left(1 - \frac{\tau_s}{\tau_1} - \frac{2}{3}\frac{\tau_c}{\tau_1}\right), \sqrt{3}\left(1 - \frac{\tau_s}{\tau_1} - \frac{2}{3}\frac{\tau_c}{\tau_1}\right), 1 - \frac{\tau_s}{\tau_1} - 2\frac{\tau_c}{\tau_1}, 1 - \frac{\tau_s}{\tau_1} - 2\frac{\tau_c}{\tau_1}, 2, 2\right]\right).$$

$$D^{-\frac{1}{2}}AD^{-\frac{1}{2}} \approx \widehat{D^{-\frac{1}{2}}}\widehat{A}\widehat{D^{-\frac{1}{2}}}$$

$$= \begin{bmatrix} \frac{2}{3}\left(1 - \frac{\tau_s}{\tau_1} - \frac{4}{3}\frac{\tau_c}{\tau_1}\right) & \frac{1}{3}\left(1 + 2\frac{\tau_s}{\tau_1} - \frac{4}{3}\frac{\tau_c}{\tau_1}\right) & \sqrt{3}\frac{\tau_c}{\tau_1} & \frac{1}{\sqrt{3}}\frac{\tau_c}{\tau_1} & 0 & 0 \\ \frac{1}{3}\left(1 + 2\frac{\tau_s}{\tau_1} - \frac{4}{3}\frac{\tau_c}{\tau_1}\right) & \frac{2}{3}\left(1 - \frac{\tau_s}{\tau_1} - \frac{4}{3}\frac{\tau_c}{\tau_1}\right) & \frac{1}{\sqrt{3}}\frac{\tau_c}{\tau_1} & \sqrt{3}\frac{\tau_c}{\tau_1} & 0 & 0 \\ \sqrt{3}\frac{\tau_c}{\tau_1} & \frac{1}{\sqrt{3}}\frac{\tau_c}{\tau_1} & 1 - 2\frac{\tau_s}{\tau_1} - 4\frac{\tau_c}{\tau_1} & 2\frac{\tau_s}{\tau_1} & 0 & 0 \\ \frac{1}{\sqrt{3}}\frac{\tau_c}{\tau_1} & \sqrt{3}\frac{\tau_c}{\tau_1} & 2\frac{\tau_s}{\tau_1} & 1 - 2\frac{\tau_s}{\tau_1} - 4\frac{\tau_c}{\tau_1} & 0 & 0 \\ 0 & 0 & 0 & 0 & \frac{1}{2} & \frac{1}{2} \\ 0 & 0 & 0 & 0 & \frac{1}{2} & \frac{1}{2} \end{bmatrix}.$$

And we have

$$\left\|D^{-\frac{1}{2}}AD^{-\frac{1}{2}} - \widehat{D^{-\frac{1}{2}}}\widehat{A}\widehat{D^{-\frac{1}{2}}}\right\|_2$$
$$\leq \left\|D^{-\frac{1}{2}}AD^{-\frac{1}{2}} - \widehat{D^{-\frac{1}{2}}}\widehat{A}\widehat{D^{-\frac{1}{2}}}\right\|_F$$
$$\leq O((\frac{\tau_c}{\tau_1})^2).$$

Let $\hat{\lambda}_a, \ldots, \hat{\lambda}_f$ be six eigenvalues of $\widehat{D^{-\frac{1}{2}}}\widehat{A}\widehat{D^{-\frac{1}{2}}}$, and $\hat{v}_a, \ldots, \hat{v}_f$ be corresponding eigenvectors. By direct calculation we have

$$\hat{\lambda}_a = 1, \ \hat{\lambda}_b = 1, \ \hat{\lambda}_c = 1 - \frac{16}{3}\frac{\tau_c}{\tau_1}, \ \hat{\lambda}_d = 0$$

and corresponding eigenvectors as

$$\hat{v}_a = [0, 0, 0, 0, 1, 1],$$
$$\hat{v}_b = [\sqrt{3}, \sqrt{3}, 1, 1, 0, 0],$$
$$\hat{v}_c = [1, 1, -\sqrt{3}, -\sqrt{3}, 0, 0],$$
$$\hat{v}_d = [0, 0, 0, 0, 1, -1].$$

For the remaining two eigenvectors, by the symmetric property, they have the formula

$$\hat{v}_e = [\alpha(\frac{\tau_s}{\tau_1}, \frac{\tau_c}{\tau_1}), -\alpha(\frac{\tau_s}{\tau_1}, \frac{\tau_c}{\tau_1}), \beta(\frac{\tau_s}{\tau_1}, \frac{\tau_c}{\tau_1}), -\beta(\frac{\tau_s}{\tau_1}, \frac{\tau_c}{\tau_1}), 0, 0],$$
$$\hat{v}_f = [\beta(\frac{\tau_s}{\tau_1}, \frac{\tau_c}{\tau_1}), -\beta(\frac{\tau_s}{\tau_1}, \frac{\tau_c}{\tau_1}), -\alpha(\frac{\tau_s}{\tau_1}, \frac{\tau_c}{\tau_1}), \alpha(\frac{\tau_s}{\tau_1}, \frac{\tau_c}{\tau_1}), 0, 0],$$

where $\alpha, \beta$ are some real functions. Then, by solving

$$\widehat{D^{-\frac{1}{2}}\widehat{A}D^{-\frac{1}{2}}}\hat{v}_e = \hat{\lambda}_e\hat{v}_e$$

$$\widehat{D^{-\frac{1}{2}}\widehat{A}D^{-\frac{1}{2}}}\hat{v}_f = \hat{\lambda}_f\hat{v}_f,$$

we get

$$\hat{\lambda}_e = \frac{1}{9}\left(\sqrt{(3 - 12\frac{\tau_s}{\tau_1} - 16\frac{\tau_c}{\tau_1})^2 + 108(\frac{\tau_c}{\tau_1})^2} - 24\frac{\tau_s}{\tau_1} - 20\frac{\tau_c}{\tau_1} + 6\right)$$

$$\hat{\lambda}_f = \frac{1}{9}\left(-\sqrt{(3 - 12\frac{\tau_s}{\tau_1} - 16\frac{\tau_c}{\tau_1})^2 + 108(\frac{\tau_c}{\tau_1})^2} - 24\frac{\tau_s}{\tau_1} - 20\frac{\tau_c}{\tau_1} + 6\right).$$

Now, we show that $\hat{\lambda}_c > \hat{\lambda}_e$. By $\frac{\tau_c}{\tau_1} \ll 1$ and $\frac{4}{9}\tau_c \le \tau_s \le \tau_c$

$$\hat{\lambda}_c \ge \hat{\lambda}_e \Leftrightarrow 3 + 24\frac{\tau_s}{\tau_1} - 28\frac{\tau_c}{\tau_1} \ge \sqrt{(3 - 12\frac{\tau_s}{\tau_1} - 16\frac{\tau_c}{\tau_1})^2 + 108(\frac{\tau_c}{\tau_1})^2}$$

$$\Leftrightarrow 36(\frac{\tau_s}{\tau_1})^2 + 35(\frac{\tau_c}{\tau_1})^2 - 144\frac{\tau_s}{\tau_1}\frac{\tau_c}{\tau_1} + 18\frac{\tau_s}{\tau_1} - 6\frac{\tau_c}{\tau_1} \ge 0.$$

Thus, we have $1 = \hat{\lambda}_a = \hat{\lambda}_b > \hat{\lambda}_c > \hat{\lambda}_e > \hat{\lambda}_f > \hat{\lambda}_d = 0$. Moreover, we also have

$$\hat{\lambda}_c - \hat{\lambda}_e = 1 - \frac{16}{3}\frac{\tau_c}{\tau_1} - \frac{1}{9}\left(\sqrt{(3 - 12\frac{\tau_s}{\tau_1} - 16\frac{\tau_c}{\tau_1})^2 + 108(\frac{\tau_c}{\tau_1})^2} - 24\frac{\tau_s}{\tau_1} - 20\frac{\tau_c}{\tau_1} + 6\right)$$

$$\ge \Omega\left(\frac{\tau_c}{\tau_1}\right).$$

Let $\hat{\lambda}_1 = \hat{\lambda}_a, \hat{\lambda}_2 = \hat{\lambda}_b, \hat{\lambda}_3 = \hat{\lambda}_c$. Then, by Weyl's Theorem, for $i \in \{1, 2, 3\}$, we have

$$|\lambda_i - \hat{\lambda}_i| \le \left\|D^{-\frac{1}{2}}AD^{-\frac{1}{2}} - \widehat{D^{-\frac{1}{2}}\widehat{A}D^{-\frac{1}{2}}}\right\|_2 \le O((\frac{\tau_c}{\tau_1})^2).$$

By Davis-Kahan theorem, we have

$$\|\sin(U, \hat{U})\|_F \le \frac{O((\frac{\tau_c}{\tau_1})^2)}{\Omega\left(\frac{\tau_c}{\tau_1}\right)} \le O(\frac{\tau_c}{\tau_1}).$$

We finish the proof. $\qquad\qquad\qquad\qquad\qquad\qquad\qquad\qquad\qquad\qquad\qquad\qquad\square$

**Theorem B.2.** *Recall $\eta_u A^{(u)}$ is defined in Theorem B.1. Assume $1 \gg \frac{\tau_c}{\tau_1} > \frac{\tau_s}{\tau_1} > 0$ and $\tau_1 + \tau_c + \tau_s = 1$. Let $\lambda_1^{(u)}, \lambda_2^{(u)}, \lambda_3^{(u)}$ and $v_1^{(u)}, v_2^{(u)}, v_3^{(u)}$ be the largest three eigenvalues and their corresponding eigenvectors of $D^{(u)-\frac{1}{2}}(\eta_u A^{(u)})D^{(u)-\frac{1}{2}}$, which is the normalized adjacency matrix of $\eta_u A^{(u)}$. Let*

$$\hat{\lambda}_1^{(u)} = 1, \quad \hat{\lambda}_2^{(u)} = 1, \quad \hat{\lambda}_3^{(u)} = 1 - 4\frac{\tau_s}{\tau_1},$$

$$\hat{v}_1^{(u)} = [0, 0, 0, 0, 1, 1],$$

$$\hat{v}_2^{(u)} = [1, 1, 1, 1, 0, 0],$$

$$\hat{v}_3^{(u)} = [1, -1, 1, -1, 0, 0].$$

*Let $U^{(u)} = [v_1^{(u)}, v_2^{(u)}, v_3^{(u)}], \hat{U}^{(u)} = [\hat{v}_1^{(u)}, \hat{v}_2^{(u)}, \hat{v}_3^{(u)}]$. Then, for $i \in \{1, 2, 3\}$, we have $|\lambda_i^{(u)} - \hat{\lambda}_i^{(u)}| \le O((\frac{\tau_c}{\tau_1})^2)$ and $\|\sin(U^{(u)}, \hat{U}^{(u)})\|_F \le O(\frac{\tau_c^2}{\tau_1(\tau_c - \tau_s)})$.*

*Proof.* Similar to the proof of Theorem B.1, up to error $O((\frac{\tau_c}{\tau_1})^2)$, we have the following equation,

$$\widehat{\eta_u A^{(u)}} = \tau_1^2 \begin{bmatrix} 1 & 2\frac{\tau_s}{\tau_1} & 2\frac{\tau_c}{\tau_1} & 0 & 0 & 0 \\ 2\frac{\tau_s}{\tau_1} & 1 & 0 & 2\frac{\tau_c}{\tau_1} & 0 & 0 \\ 2\frac{\tau_c}{\tau_1} & 0 & 1 & 2\frac{\tau_s}{\tau_1} & 0 & 0 \\ 0 & 2\frac{\tau_c}{\tau_1} & 2\frac{\tau_s}{\tau_1} & 1 & 0 & 0 \\ 0 & 0 & 0 & 0 & 2 & 2 \\ 0 & 0 & 0 & 0 & 2 & 2 \end{bmatrix}.$$

$$\widehat{D^{(u)}} = \tau_1^2 diag\left(\left[1 + 2\frac{\tau_s}{\tau_1} + 2\frac{\tau_c}{\tau_1}, 1 + 2\frac{\tau_s}{\tau_1} + 2\frac{\tau_c}{\tau_1}, 1 + 2\frac{\tau_s}{\tau_1} + 2\frac{\tau_c}{\tau_1}, 1 + 2\frac{\tau_s}{\tau_1} + 2\frac{\tau_c}{\tau_1}, 4, 4\right]\right).$$

$$\widehat{D^{(u)-\frac{1}{2}}} = \frac{1}{\tau_1} diag\left(\left[1 - \frac{\tau_s}{\tau_1} - \frac{\tau_c}{\tau_1}, 1 - \frac{\tau_s}{\tau_1} - \frac{\tau_c}{\tau_1}, 1 - \frac{\tau_s}{\tau_1} - \frac{\tau_c}{\tau_1}, 1 - \frac{\tau_s}{\tau_1} - \frac{\tau_c}{\tau_1}, 2, 2\right]\right).$$

$$\widehat{D^{(u)-\frac{1}{2}}}\widehat{\eta_u A^{(u)}}\widehat{D^{(u)-\frac{1}{2}}} = \begin{bmatrix} 1 - 2\frac{\tau_s}{\tau_1} - 2\frac{\tau_c}{\tau_1} & 2\frac{\tau_s}{\tau_1} & 2\frac{\tau_c}{\tau_1} & 0 & 0 & 0 \\ 2\frac{\tau_s}{\tau_1} & 1 - 2\frac{\tau_s}{\tau_1} - 2\frac{\tau_c}{\tau_1} & 0 & 2\frac{\tau_c}{\tau_1} & 0 & 0 \\ 2\frac{\tau_c}{\tau_1} & 0 & 1 - 2\frac{\tau_s}{\tau_1} - 2\frac{\tau_c}{\tau_1} & 2\frac{\tau_s}{\tau_1} & 0 & 0 \\ 0 & 2\frac{\tau_c}{\tau_1} & 2\frac{\tau_s}{\tau_1} & 1 - 2\frac{\tau_s}{\tau_1} - 2\frac{\tau_c}{\tau_1} & 0 & 0 \\ 0 & 0 & 0 & 0 & \frac{1}{2} & \frac{1}{2} \\ 0 & 0 & 0 & 0 & \frac{1}{2} & \frac{1}{2} \end{bmatrix}.$$

Let $\hat{\lambda}_1^{(u)}, \ldots, \hat{\lambda}_6^{(u)}$ be six eigenvalue of $\widehat{D^{(u)-\frac{1}{2}}}\widehat{\eta_u A^{(u)}}\widehat{D^{(u)-\frac{1}{2}}}$, and $\hat{v}_1^{(u)}, \ldots, \hat{v}_6^{(u)}$ be corresponding eigenvectors. By direct calculation we have

$$\hat{\lambda}_1^{(u)} = 1, \quad \hat{\lambda}_2^{(u)} = 1, \quad \hat{\lambda}_3^{(u)} = 1 - 4\frac{\tau_s}{\tau_1}, \quad \hat{\lambda}_4^{(u)} = 1 - 4\frac{\tau_c}{\tau_1}, \quad \hat{\lambda}_5^{(u)} = 1 - 4\frac{\tau_s}{\tau_1} - 4\frac{\tau_c}{\tau_1}, \quad \hat{\lambda}_6^{(u)} = 0$$

and corresponding eigenvector as

$$\hat{v}_1^{(u)} = [0, 0, 0, 0, 1, 1],$$
$$\hat{v}_2^{(u)} = [1, 1, 1, 1, 0, 0],$$
$$\hat{v}_3^{(u)} = [1, -1, 1, -1, 0, 0],$$
$$\hat{v}_4^{(u)} = [1, 1, -1, -1, 0, 0],$$
$$\hat{v}_5^{(u)} = [1, -1, -1, 1, 0, 0],$$
$$\hat{v}_6^{(u)} = [0, 0, 0, 0, 1, -1].$$

Then, by Weyl's Theorem, for $i \in \{1, 2, 3\}$, we have

$$|\lambda_i^{(u)} - \hat{\lambda}_i^{(u)}| \leq \left\|D^{(u)-\frac{1}{2}}\eta_u A^{(u)} D^{(u)-\frac{1}{2}} - \widehat{D^{(u)-\frac{1}{2}}}\widehat{\eta_u A^{(u)}}\widehat{D^{(u)-\frac{1}{2}}}\right\|_2 \leq O\left(\left(\frac{\tau_c}{\tau_1}\right)^2\right).$$

By Davis-Kahan theorem, we have

$$\|\sin(U^{(u)}, \hat{U}^{(u)})\|_F \leq \frac{O\left(\left(\frac{\tau_c}{\tau_1}\right)^2\right)}{4\left(\frac{\tau_c}{\tau_1} - \frac{\tau_s}{\tau_1}\right)} \leq O\left(\frac{\tau_c^2}{\tau_1(\tau_c - \tau_s)}\right).$$

We finish the proof. $\qquad\square$

# C  Technical Details for Main Theory

## C.1  Notation

We let $\mathbf{1}_n$, $\mathbf{0}_n$ be the $n$-dimensional vector with all 1 or 0 values respectively. $\mathbf{1}_{m\times n}$, $\mathbf{0}_{m\times n}$ are similarly defined for $m$-by-$n$ matrix. $I_n$ is the identity matrix with shape $n \times n$. For any matrix $V$, $V_{(i,j)}$ indictes the value at $i$-th row and $j$-th column of $V$. If the matrix is subscripted like $V_k$, we use a comma in-between like $V_{k,(i,j)}$. Similarly, $\mathbf{v}_{(i)}$ and $\mathbf{v}_{k,(i)}$ are the $i$-th value for vector $\mathbf{v}$ and $\mathbf{v}_k$ respectively. $[n]$ is used to abbreviate the set $\{1, 2, ..., n\}$.

## C.2  Matrix Form of K-means and the Derivative

Recall that we defined the K-means clustering measure of features in Sec. 4:

$$\mathcal{M}_{kms}(\Pi, Z) = \sum_{\pi\in\Pi}\sum_{i\in\pi}\|\mathbf{z}_i - \boldsymbol{\mu}_\pi\|^2 \Big/ \sum_{\pi\in\Pi}|\pi|\,\|\boldsymbol{\mu}_\pi - \boldsymbol{\mu}_\Pi\|^2 , \tag{12}$$

where the numerator measures the intra-class distance:

$$\mathcal{M}_{intra}(\Pi, Z) = \sum_{\pi\in\Pi}\sum_{i\in\pi}\|\mathbf{z}_i - \boldsymbol{\mu}_\pi\|^2 , \tag{13}$$

and the denominator measures the inter-class distance:

$$\mathcal{M}_{inter}(\Pi, Z) = \sum_{\pi\in\Pi}|\pi|\,\|\boldsymbol{\mu}_\pi - \boldsymbol{\mu}_\Pi\|^2 . \tag{14}$$

We will show next how to convert the intra-class and the inter-class measure into a matrix form, which is desirable for analysis.

**Intra-class measure.**  Note that the $K$-means intra-class measure can be rewritten in a matrix form:

$$\mathcal{M}_{intra}(\Pi, Z) = \|Z - H_\Pi Z\|_F^2,$$

where $H_\Pi$ is a matrix to convert $Z$ to mean vectors w.r.t clusters defined by $\Pi$. Without losing the generality, we assume $Z$ is ordered according to the partition in $\Pi$ — first $|\pi_1|$ vectors are in $\pi_1$, next $|\pi_2|$ vectors are in $\pi_2$, etc. Then $H_\Pi$ is given by:

$$H_\Pi = \begin{bmatrix} \frac{1}{|\pi_1|}\mathbf{1}_{|\pi_1|\times|\pi_1|} & \mathbf{0} & ... & \mathbf{0} \\ \mathbf{0} & \frac{1}{|\pi_2|}\mathbf{1}_{|\pi_2|\times|\pi_2|} & ... & \mathbf{0} \\ ... & ... & ... & ... \\ \mathbf{0} & \mathbf{0} & ... & \frac{1}{|\pi_k|}\mathbf{1}_{|\pi_k|\times|\pi_k|} \end{bmatrix}.$$

Going further, we have:

$$\begin{aligned} \mathcal{M}_{intra}(\Pi, Z) &= \|Z - H_\Pi Z\|_F^2 \\ &= \mathrm{Tr}((I - H_\Pi)^2 Z Z^\top) \\ &= \mathrm{Tr}((I - 2H_\Pi + H_\Pi^2) Z Z^\top) \\ &= \mathrm{Tr}((I - H_\Pi) Z Z^\top). \end{aligned}$$

**Inter-class measure.**  The inter-class measure can be equivalently given by:

$$\mathcal{M}_{inter}(\Pi, Z) = \|H_\Pi Z - \frac{1}{N}\mathbf{1}_{N\times N} Z\|_F^2,$$

where $H_\Pi$ is defined as above. And we can also derive:

$$\begin{aligned} \mathcal{M}_{inter}(\Pi, Z) &= \|H_\Pi Z - \frac{1}{N}\mathbf{1}_{N\times N} Z\|_F^2 \\ &= \mathrm{Tr}((H_\Pi - \frac{1}{N}\mathbf{1}_{N\times N})^2 Z Z^\top) \\ &= \mathrm{Tr}((H_\Pi^2 - \frac{2}{N}H_\Pi\mathbf{1}_{N\times N} + \frac{1}{N^2}\mathbf{1}_{N\times N}^2) Z Z^\top) \\ &= \mathrm{Tr}((H_\Pi - \frac{1}{N}\mathbf{1}_{N\times N}) Z Z^\top). \end{aligned}$$

## C.3 K-means Measure Has the Same Order as K-means Error

**Theorem C.1.** *(Recap of Theorem 4.1) We define the $\xi_{\pi \to \pi'}$ as the index of samples that is from class division $\pi$ however is closer to $\boldsymbol{\mu}_{\pi'}$ than $\boldsymbol{\mu}_\pi$. In other word, $\xi_{\pi \to \pi'} = \{i : i \in \pi, \|\mathbf{z}_i - \boldsymbol{\mu}_\pi\|_2 \geq \|\mathbf{z}_i - \boldsymbol{\mu}_{\pi'}\|_2\}$. Assuming $|\xi_{\pi \to \pi'}| > 0$, we define below the clustering error ratio from $\pi$ to $\pi'$ as $\mathcal{E}_{\pi \to \pi'}$ and the overall cluster error ratio $\mathcal{E}_{\Pi,Z}$ as the **Harmonic Mean** of $\mathcal{E}_{\pi \to \pi'}$ among all class pairs:*

$$\mathcal{E}_{\Pi,Z} = C(C-1) / \left( \sum_{\substack{\pi \neq \pi' \\ \pi, \pi' \in \Pi}} \frac{1}{\mathcal{E}_{\pi \to \pi'}} \right), \text{ where } \mathcal{E}_{\pi \to \pi'} = \frac{|\xi_{\pi \to \pi'}|}{|\pi'| + |\pi|}.$$

*The K-means measure $\mathcal{M}_{kms}(\Pi, Z)$ has the same order as the Harmonic Mean of the cluster error ratio between all cluster pairs:*

$$\mathcal{E}_{\Pi,Z} = O(\mathcal{M}_{kms}(\Pi, Z)).$$

*Proof.* We have the following inequality for $i \in \xi_{\pi \to \pi'}$:

$$4\|\mathbf{z}_i - \boldsymbol{\mu}_\pi\|_2^2 \geq 2\|\mathbf{z}_i - \boldsymbol{\mu}_\pi\|_2^2 + 2\|\mathbf{z}_i - \boldsymbol{\mu}_{\pi'}\|_2^2 \geq \|\boldsymbol{\mu}_\pi - \boldsymbol{\mu}_{\pi'}\|_2^2.$$

Then we have:

$$\begin{aligned}
\mathcal{M}_{intra}(\Pi, Z) &= \sum_{\pi \in \Pi} \sum_{i \in \pi} \|\mathbf{z}_i - \boldsymbol{\mu}_\pi\|_2^2 \\
&\geq \sum_{i \in \pi} \|\mathbf{z}_i - \boldsymbol{\mu}_\pi\|_2^2 \\
&\geq \sum_{i \in \xi_{\pi \to \pi'}} \|\mathbf{z}_i - \boldsymbol{\mu}_\pi\|_2^2 \\
&\geq \frac{1}{4} \sum_{i \in \xi_{\pi \to \pi'}} \|\boldsymbol{\mu}_\pi - \boldsymbol{\mu}_{\pi'}\|_2^2 \\
&= \frac{1}{4} |\xi_{\pi \to \pi'}| \|\boldsymbol{\mu}_\pi - \boldsymbol{\mu}_{\pi'}\|_2^2.
\end{aligned}$$

Note that the inter-class measure can be decomposed into the summation of cluster center distances:

$$\begin{aligned}
\mathcal{M}_{inter}(\Pi, Z) &= \sum_{\pi \in \Pi} |\pi| \|\boldsymbol{\mu}_\pi - \boldsymbol{\mu}_\Pi\|_2^2 \\
&= \sum_{\pi \in \Pi} \frac{|\pi|}{N^2} \left\| \left(\sum_{\pi' \in \Pi} |\pi'|\right) \boldsymbol{\mu}_\pi - \sum_{\pi' \in \Pi} |\pi'| \boldsymbol{\mu}_{\pi'} \right\|_2^2 \\
&\leq \frac{C}{N^2} \sum_{\pi \in \Pi} |\pi| \sum_{\pi' \in \Pi} |\pi'|^2 \|\boldsymbol{\mu}_\pi - \boldsymbol{\mu}_{\pi'}\|_2^2 \\
&= \frac{C}{N^2} \sum_{\pi \neq \pi'} |\pi||\pi'|(|\pi'| + |\pi|) \|\boldsymbol{\mu}_\pi - \boldsymbol{\mu}_{\pi'}\|_2^2,
\end{aligned}$$

where $\sum_{\pi \neq \pi'}$ is enumerating over any two different class partitions in $\Pi$. Combining together, we have:

$$\begin{aligned}
C(C-1) / \left( \sum_{\pi \neq \pi'} \frac{(|\pi'| + |\pi|)}{|\xi_{\pi \to \pi'}|} \right) &= C(C-1) / \left( \sum_{\pi \neq \pi'} \frac{(|\pi'| + |\pi|)\|\boldsymbol{\mu}_\pi - \boldsymbol{\mu}_{\pi'}\|_2^2}{|\xi_{\pi \to \pi'}|\|\boldsymbol{\mu}_\pi - \boldsymbol{\mu}_{\pi'}\|_2^2} \right) \\
&\leq C(C-1) / \left( \sum_{\pi \neq \pi'} \frac{|\pi'||\pi|(|\pi'| + |\pi|)\|\boldsymbol{\mu}_\pi - \boldsymbol{\mu}_{\pi'}\|_2^2}{N^2|\xi_{\pi \to \pi'}|\|\boldsymbol{\mu}_\pi - \boldsymbol{\mu}_{\pi'}\|_2^2} \right) \\
&\leq C(C-1) / \left( \frac{\mathcal{M}_{inter}(\Pi, Z)}{4C\mathcal{M}_{intra}(\Pi, Z)} \right) \\
&= O(\mathcal{M}_{kms}(\Pi, Z)).
\end{aligned}$$

$\square$

## C.4 Proof of Theorem 4.2

We start by providing more details to supplement Sec. 4.2.1.

**Matrix perturbation by adding labels.** Recall that we define in Eq. 3 that the adjacency matrix is the unlabeled one $A^{(u)}$ plus the perturbation of the label information $A^{(l)}$:

$$A = \eta_u A^{(u)} + \eta_l A^{(l)}.$$

We study the perturbation from two aspects: (1) The direction of the perturbation which is given by $A^{(l)}$, (2) The perturbation magnitude $\eta_l$. We first consider the perturbation direction $A^{(l)}$ and recall that we defined the concrete form in Eq. 2:

$$A^{(l)}_{xx'} = w^{(l)}_{xx'} \triangleq \sum_{i \in \mathcal{Y}_l} \mathbb{E}_{\bar{x}_l \sim \mathcal{P}_{l_i}} \mathbb{E}_{\bar{x}'_l \sim \mathcal{P}_{l_i}} \mathcal{T}(x|\bar{x}_l) \mathcal{T}(x'|\bar{x}'_l).$$

For simplicity, we consider $|\mathcal{Y}_l| = 1$ in this theoretical analysis. Then we observe that $A^{(l)}_{xx'}$ is a rank-1 matrix can be written as

$$A^{(l)}_{xx'} = \mathfrak{l}\mathfrak{l}^\top,$$

where $\mathfrak{l} \in \mathbb{R}^{N \times 1}$ with $(\mathfrak{l})_x = \mathbb{E}_{\bar{x}_l \sim \mathcal{P}_{l_1}} \mathcal{T}(x|\bar{x}_l)$. And we define $D_l \triangleq diag(\mathfrak{l})$.

**The perturbation function of representation.** We then consider a more generalized form for the adjacency matrix:

$$A(\delta) \triangleq \eta_u A^{(u)} + \delta \mathfrak{l}\mathfrak{l}^\top.$$

where we treat the adjacency matrix as a function of the "labeling perturbation" degree $\delta$. It is clear that $A(0) = \eta_u A^{(u)}$ which is the scaled adjacency matrix for the unlabeled case and that $A(\eta_l) = A$. When we let the adjacency matrix be a function of $\delta$, the normalized form and the derived feature representation should also be the function of $\delta$. We proceed by defining these terms.

Without losing the generality, we let $diag(\mathbf{1}_N^\top A(0)) = I_N$ which means the node in the unlabeled graph has equal degree. We then have:

$$D(\delta) \triangleq diag(\mathbf{1}_N^\top A(\delta)) = I_N + \delta D_l.$$

The normalized adjacency matrix is given by:

$$\tilde{A}(\delta) \triangleq D(\delta)^{-\frac{1}{2}} A(\delta) D(\delta)^{-\frac{1}{2}}.$$

For feature representation $Z(\delta)$, it is derived from the top-$k$ SVD components of $\tilde{A}(\delta)$. Specifically, we have:

$$Z(\delta) Z(\delta)^\top = D(\delta)^{-\frac{1}{2}} \tilde{A}_k(\delta) D(\delta)^{-\frac{1}{2}} = D(\delta)^{-\frac{1}{2}} \sum_{j=1}^k \lambda_j(\delta) \Phi_j(\delta) D(\delta)^{-\frac{1}{2}},$$

where we define $\tilde{A}_k(\delta)$ as the top-$k$ SVD components of $\tilde{A}(\delta)$ and can be further written as $\tilde{A}_k(\delta) = \sum_{j=1}^k \lambda_j(\delta) \Phi_j(\delta)$. Here the $\lambda_j(\delta)$ is the $j$-th singular value and $\Phi_j(\delta)$ is the $j$-th singular projector $(\Phi_j(\delta) = v_j(\delta) v_j(\delta)^\top)$ defined by the $j$-th singular vector $v_j(\delta)$. **For brevity, when $\delta = 0$, we remove the suffix** $(0)$ since it is equivalent to the unperturbed version of notations. For example, we let

$$\tilde{A}(0) = \tilde{A}^{(u)}, Z(0) = Z^{(u)}, \lambda_i(0) = \lambda_i^{(u)}, v_i(0) = v_i^{(u)}, \Phi_i(0) = \Phi_i^{(u)}.$$

**Theorem C.2.** *(Recap of Th. 4.2) Denote $V_\varnothing^{(u)} \in \mathbb{R}^{N \times (N-k)}$ as the null space of $V_k^{(u)}$ and $\tilde{A}_k^{(u)} = V_k^{(u)} \Sigma_k^{(u)} V_k^{(u)\top}$ as the rank-k approximation for $\tilde{A}^{(u)}$. Given $\delta, \eta_1 > 0$ and let $\mathcal{G}_k$ as the spectral gap between k-th and k + 1-th singular values of $\tilde{A}^{(u)}$, we have:*

$$\Delta_{kms}(\delta) = \delta \eta_1 \, \mathrm{Tr} \left( \Upsilon \left( V_k^{(u)} V_k^{(u)\top} \mathfrak{l}\mathfrak{l}^\top (I + V_\varnothing^{(u)} V_\varnothing^{(u)\top}) - 2\tilde{A}_k^{(u)} diag(\mathfrak{l}) \right) \right) + O(\frac{1}{\mathcal{G}_k} + \delta^2),$$

*where $diag(\cdot)$ converts the vector to the corresponding diagonal matrix and $\Upsilon \in \mathbb{R}^{N \times N}$ is a matrix encoding the **ground-truth clustering structure** in the way that $\Upsilon_{xx'} > 0$ if $x$ and $x'$ has the same label and $\Upsilon_{xx'} < 0$ otherwise.*

*Proof.* As we shown in Sec C.2, we can now also write the K-means measure as the function of perturbation:

$$\mathcal{M}_{kms}(\delta) = \frac{\text{Tr}((I - H_\Pi)Z(\delta)Z(\delta)^\top)}{\text{Tr}((H_\Pi - \frac{1}{N}\mathbf{1}_{N\times N})Z(\delta)Z(\delta)^\top)}.$$

The proof is directly given by the following Lemma C.3. □

**Lemma C.3.** *Let* $\eta_1, \eta_2$ *be two real values and* $\Upsilon = (1 + \eta_2)H_\Pi - I - \frac{\eta_2}{N}\mathbf{1}_N\mathbf{1}_N^\top$. *Let the spectrum gap* $\mathcal{G}_k = \frac{\lambda_k^{(u)}}{\lambda_{k+1}^{(u)}}$, *we have the derivative of the K-means measure evaluated at* $\delta = 0$:

$$[\mathcal{M}_{kms}(\delta)]'\Big|_{\delta=0} = -\eta_1 \text{Tr}\left(\Upsilon\left(V_k^{(u)}V_k^{(u)\top}\mathfrak{l}\mathfrak{l}^\top - 2\tilde{A}_k^{(u)}D_l + V_k^{(u)}V_k^{(u)\top}\mathfrak{l}\mathfrak{l}^\top V_\varnothing^{(u)}V_\varnothing^{(u)\top}\right)\right) + O(\frac{1}{\mathcal{G}_k}).$$

The proof for Lemma C.3 is lengthy. We postpone it to Sec. C.6.

## C.5 Proof of Theorem 4.3

We start by showing the justification of the assumptions made in Theorem 4.3.

**Assumption C.4.** We assume the spectral gap $\mathcal{G}_k$ is large. Such an assumption is commonly used in theory works using spectral analysis [32, 57].

**Assumption C.5.** We assume $\mathfrak{l}$ lies in the linear span of $V_k^{(u)}$. *i.e.,* $V_k^{(u)}V_k^{(u)\top}\mathfrak{l} = \mathfrak{l}, V_\varnothing^{(u)\top}\mathfrak{l} = 0$. The goal of this assumption is to simplify $(V_k^{(u)}V_k^{(u)\top}\mathfrak{l}\mathfrak{l}^\top + V_k^{(u)}V_k^{(u)\top}\mathfrak{l}\mathfrak{l}^\top V_\varnothing^{(u)}V_\varnothing^{(u)\top})$ to $\mathfrak{l}\mathfrak{l}^\top$.

**Assumption C.6.** For any $\pi_c \in \Pi$, $\forall i, j \in \pi_c, \mathfrak{l}_{(i)} = \mathfrak{l}_{(j)} =: \mathfrak{l}_{\pi_c}$. Recall that the $\mathfrak{l}_{(i)}$ means the connection between the $i$-th sample to the labeled data. Here we can view $\mathfrak{l}_{\pi_c}$ as the *connection between class $c$ to the labeled data.*

**Theorem C.7.** *(Recap of Theorem 4.3.)* With Assumption C.4, C.5 and C.6. Given $\delta, \eta_1, \eta_2 > 0$, we have:

$$\Delta_{kms}(\delta) \geq \delta\eta_1\eta_2 \sum_{\pi_c \in \Pi} |\pi_c|\mathfrak{l}_{\pi_c}\Delta_{\pi_c}(\delta),$$

*where*

$$\Delta_{\pi_c}(\delta) = (\mathfrak{l}_{\pi_c} - \frac{1}{N}) - 2(1 - \frac{|\pi_c|}{N})(\mathbb{E}_{i\in\pi_c}\mathbb{E}_{j\in\pi_c}\mathbf{z}_i^\top\mathbf{z}_j - \mathbb{E}_{i\in\pi_c}\mathbb{E}_{j\notin\pi_c}\mathbf{z}_i^\top\mathbf{z}_j).$$

*Proof.* The proof is directly given by Lemma C.8 and plugging the definition of $\Delta_{kms}(\delta)$. □

**Lemma C.8.** *With Assumption C.4 C.5 and C.6, we have the derivative of K-means measure with the upper bound:*

$$[\mathcal{M}_{kms}(\delta)]'\Big|_{\delta=0} \leq -\eta_1\eta_2 \sum_{\pi\in\Pi} |\pi|\mathfrak{l}_\pi\left((\mathfrak{l}_\pi - \frac{1}{N}) - 2(\boldsymbol{\mu}_\pi^\top\boldsymbol{\mu}_\pi - \boldsymbol{\mu}_\pi^\top\boldsymbol{\mu}_\Pi)\right).$$

*Proof.* By Assumption C.4 C.5 and C.6 and Theorem 4.2, we have

$$\begin{aligned}
\frac{1}{\eta_1}[\mathcal{M}_{kms}(\delta)]'\Big|_{\delta=0} &= -\text{Tr}\left(\Upsilon\left(V_k^{(u)}V_k^{(u)\top}\mathfrak{l}\mathfrak{l}^\top - 2\tilde{A}_k^{(u)}D_l\right)\right) \\
&= -\text{Tr}\left(\Upsilon\left(\mathfrak{l}\mathfrak{l}^\top - 2\tilde{A}_k^{(u)}D_l\right)\right) \\
&= -\text{Tr}\left(\left((1+\eta_2)H_\Pi - I - \frac{\eta_2}{N}\mathbf{1}_N\mathbf{1}_N^\top\right)\left(\mathfrak{l}\mathfrak{l}^\top - 2\tilde{A}_k^{(u)}D_l\right)\right) \\
&= (1+\eta_2)\mathcal{M}'_H + \mathcal{M}'_I + \eta_2\mathcal{M}'_{\mathbf{1}},
\end{aligned}$$

where

$$\mathcal{M}'_H = -\operatorname{Tr}\left(H_\Pi\left(\mathbb{1}^\top - 2\tilde{A}_k^{(u)}D_l\right)\right)$$

$$= -\sum_{\pi\in\Pi}\left(|\pi|(\mathbb{E}_{i\in\pi}\mathfrak{l}_{(i)})^2 - \frac{2}{|\pi|}\sum_{i\in\pi}\sum_{j\in\pi}\mathfrak{l}_{(i)}\tilde{A}_{k,(i,j)}^{(u)}\right)$$

$$= -\sum_{\pi\in\Pi}\left(|\pi|\mathfrak{l}_\pi^2 - 2|\pi|\mathfrak{l}_\pi\mathbb{E}_{(i,j)\in\pi\times\pi}\mathbf{z}_i^\top\mathbf{z}_j\right)$$

$$= -\sum_{\pi\in\Pi}|\pi|\mathfrak{l}_\pi(\mathfrak{l}_\pi - 2\boldsymbol{\mu}_\pi^\top\boldsymbol{\mu}_\pi),$$

$$\mathcal{M}'_I = \operatorname{Tr}\left(\left(\mathbb{1}^\top - 2\tilde{A}_k^{(u)}D_l\right)\right)$$

$$= \sum_{\pi\in\Pi}|\pi|\mathfrak{l}_\pi(\mathfrak{l}_\pi - 2\mathbb{E}_{i\in\pi}\mathbf{z}_i^\top\mathbf{z}_i),$$

and

$$\mathcal{M}'_{\mathbf{1}} = \operatorname{Tr}\left(\frac{1}{N}\mathbf{1}_N\mathbf{1}_N^\top\left(\mathbb{1}^\top - 2\tilde{A}_k^{(u)}D_l\right)\right)$$

$$= \frac{1}{N} - 2\sum_{\pi\in\Pi}\sum_{i\in\pi}\mathfrak{l}_{(i)}\mathbb{E}_{j\in[N]}\mathbf{z}_i^\top\mathbf{z}_j$$

$$= \frac{1}{N} - 2\sum_{\pi\in\Pi}|\pi|\mathfrak{l}_\pi\boldsymbol{\mu}_\pi^\top\boldsymbol{\mu}_\Pi.$$

We observe that

$$\mathcal{M}'_I + \mathcal{M}'_H = -\sum_{\pi\in\Pi}|\pi|\mathfrak{l}_\pi(\mathfrak{l}_\pi - 2\boldsymbol{\mu}_\pi^\top\boldsymbol{\mu}_\pi) + \sum_{\pi\in\Pi}|\pi|\mathfrak{l}_\pi(\mathfrak{l}_\pi - 2\mathbb{E}_{i\in\pi}\mathbf{z}_i^\top\mathbf{z}_i)$$

$$= 2\sum_{\pi\in\Pi}|\pi|\mathfrak{l}_\pi(\|\mathbb{E}_{i\in\pi}\mathbf{z}_i\|_2^2 - \mathbb{E}_{i\in\pi}\|\mathbf{z}_i\|_2^2)$$

$$\le 0,$$

where the last inequality is by Jensen's Inequality. We then have

$$\frac{1}{\eta_1\eta_2}[\mathcal{M}_{kms}(\delta)]'\Big|_{\delta=0} \le \mathcal{M}'_H + \mathcal{M}'_{\mathbf{1}}$$

$$= -\sum_{\pi\in\Pi}|\pi|\mathfrak{l}_\pi(\mathfrak{l}_\pi - 2\boldsymbol{\mu}_\pi^\top\boldsymbol{\mu}_\pi) + \frac{1}{N} - 2\sum_{\pi\in\Pi}|\pi|\mathfrak{l}_\pi\boldsymbol{\mu}_\pi^\top\boldsymbol{\mu}_\Pi$$

$$= \frac{1}{N} - \sum_{\pi\in\Pi}|\pi|\mathfrak{l}_\pi(\mathfrak{l}_\pi - 2(\boldsymbol{\mu}_\pi^\top\boldsymbol{\mu}_\pi - \boldsymbol{\mu}_\pi^\top\boldsymbol{\mu}_\Pi))$$

$$= -\sum_{\pi\in\Pi}|\pi|\mathfrak{l}_\pi((\mathfrak{l}_\pi - \frac{1}{N}) - 2(\boldsymbol{\mu}_\pi^\top\boldsymbol{\mu}_\pi - \boldsymbol{\mu}_\pi^\top\boldsymbol{\mu}_\Pi)).$$

$\square$

## C.6   Proof of Lemma C.3

**Notation Recap:** We define $\tilde{A}_k(\delta)$ as the top-$k$ SVD components of $\tilde{A}(\delta)$ and can be further written as $\tilde{A}_k(\delta) = \sum_{j=1}^k \lambda_j(\delta)\Phi_j(\delta)$. Here the $\lambda_j(\delta)$ is the $j$-th singular value and $\Phi_j(\delta)$ is the $j$-th singular projector ($\Phi_j(\delta) = v_j(\delta)v_j(\delta)^\top$) defined by the $j$-th singular vector $v_j(\delta)$. **For brevity, when $\delta = 0$, we remove the suffix** $(0)$ since it is equivalent to the unperturbed version of notations. For example, we let

$$\tilde{A}(0) = \tilde{A}^{(u)}, Z(0) = Z^{(u)}, \lambda_i(0) = \lambda_i^{(u)}, v_i(0) = v_i^{(u)}, \Phi_i(0) = \Phi_i^{(u)}.$$

*Proof.* By the derivative rule, we have,

$$
\begin{aligned}
\mathcal{M}'_{kms}(\delta) &= \frac{1}{\mathcal{M}_{inter}(\Pi, Z)} \mathcal{M}'_{intra}(\delta) - \frac{\mathcal{M}_{intra}(\Pi, Z)}{\mathcal{M}_{inter}(\Pi, Z)^2} \mathcal{M}'_{inter}(\delta) \\
&= \eta_1 \mathcal{M}'_{intra}(\delta) - \eta_1 \eta_2 \mathcal{M}'_{inter}(\delta) \\
&= \eta_1 \left( \mathrm{Tr}((I_\Pi - H_\Pi)[Z(\delta)Z(\delta)^\top]') - \eta_2 \mathrm{Tr}((H_\Pi - \frac{1}{N}\mathbf{1}_{N\times N})[Z(\delta)Z(\delta)^\top]') \right) \\
&= \eta_1 \left( \mathrm{Tr}((I_\Pi + \frac{\eta_2}{N}\mathbf{1}_{N\times N} - (\eta_2+1)H_\Pi)[Z(\delta)Z(\delta)^\top]') \right) \\
&= -\eta_1 \left( \mathrm{Tr}(\Upsilon[Z(\delta)Z(\delta)^\top]') \right) \\
&= -\eta_1 \sum_{j=1}^k \mathrm{Tr}(\Upsilon[D(\delta)^{-\frac{1}{2}}\lambda_j(\delta)\Phi_j(\delta)D(\delta)^{-\frac{1}{2}}]'),
\end{aligned}
$$

where we let $\eta_1 = \frac{1}{\mathcal{M}_{inter}(\Pi,Z)}$, $\eta_2 = \frac{\mathcal{M}_{intra}(\Pi,Z)}{\mathcal{M}_{inter}(\Pi,Z)}$ and $\Upsilon = (1+\eta_2)H_\Pi - I_\Pi - \frac{\eta_2}{N}\mathbf{1}_N\mathbf{1}_N^\top$. We proceed by showing the calculation of $[D(\delta)^{-\frac{1}{2}}]'$, $[\lambda_j(\delta)]'$ and $[\Phi_j(\delta)]'$.

Since $D(\delta) = I + \delta D_l$, then $[D(\delta)^{-\frac{1}{2}}]'\big|_{\delta=0} = -\frac{1}{2}D_l$. To calculate $[\lambda_j(\delta)]'$ and $[\Phi_j(\delta)]'$, we first need:

$$
\begin{aligned}
[\tilde{A}(\delta)]'\big|_{\delta=0} &= [D(\delta)^{-\frac{1}{2}}A(\delta)D(\delta)^{-\frac{1}{2}}]' \\
&= [D(\delta)^{-\frac{1}{2}}]'\tilde{A}^{(u)} + [A(\delta)]' + \tilde{A}^{(u)}[D(\delta)^{-\frac{1}{2}}]' \\
&= -\frac{1}{2}D_l\tilde{A}^{(u)} + \mathbb{1}^\top - \frac{1}{2}\tilde{A}^{(u)}D_l.
\end{aligned}
$$

Then, according to Equation (3) in [20], we have:

$$
\begin{aligned}
[\lambda_j(\delta)]'\big|_{\delta=0} &= \mathrm{Tr}(\Phi_j^{(u)}[\tilde{A}(\delta)]') \\
&= \mathrm{Tr}(\Phi_j^{(u)}(-\frac{1}{2}D_l\tilde{A}^{(u)} + \mathbb{1}^\top - \frac{1}{2}\tilde{A}^{(u)}D_l)) \\
&= \mathrm{Tr}((-\frac{\lambda_j^{(u)}}{2}D_l\Phi_j^{(u)} + \Phi_j^{(u)}\mathbb{1}^\top - \frac{\lambda_j^{(u)}}{2}\Phi_j^{(u)}D_l)) \\
&= \mathrm{Tr}(\Phi_j^{(u)}(\mathbb{1}^\top - \lambda_j^{(u)}D_l)).
\end{aligned}
$$

According to Equation (10) in [20], we have:

$$
\begin{aligned}
[\Phi_j(\delta)]'\big|_{\delta=0} &= (\lambda_j^{(u)}I_N - \tilde{A}^{(u)})^\dagger[\tilde{A}(\delta)]'\Phi_j^{(u)} + \Phi_j^{(u)}[\tilde{A}(\delta)]'(\lambda_j^{(u)}I_N - \tilde{A}^{(u)})^\dagger \\
&= \sum_{i\neq j}^N \frac{1}{\lambda_j^{(u)} - \lambda_i^{(u)}}(\Phi_i^{(u)}[\tilde{A}(\delta)]'\Phi_j^{(u)} + \Phi_j^{(u)}[\tilde{A}(\delta)]'\Phi_i^{(u)}) \\
&= \sum_{i\neq j}^N \frac{1}{\lambda_j^{(u)} - \lambda_i^{(u)}}(\Phi_i^{(u)}(-\frac{1}{2}D_l\tilde{A}^{(u)} + \mathbb{1}^\top - \frac{1}{2}\tilde{A}^{(u)}D_l)\Phi_j^{(u)} + \Phi_j^{(u)}(...)\Phi_i^{(u)}) \\
&= \sum_{i\neq j}^N \frac{1}{\lambda_j^{(u)} - \lambda_i^{(u)}}(\Phi_i^{(u)}(\mathbb{1}^\top - \frac{\lambda_j^{(u)} + \lambda_i^{(u)}}{2}D_l)\Phi_j^{(u)} + \Phi_j^{(u)}(\mathbb{1}^\top - \frac{\lambda_j^{(u)} + \lambda_i^{(u)}}{2}D_l)\Phi_i^{(u)}).
\end{aligned}
$$

Now we calculate the derivative of the $K$-means loss:

$$\frac{1}{\eta_1}[\mathcal{M}_{kms}(\delta)]'\Big|_{\delta=0} = -\sum_{j=1}^{k}[\mathrm{Tr}(\Upsilon D(\delta)^{-\frac{1}{2}}\lambda_j(\delta)\Phi_j(\delta)D(\delta)^{-\frac{1}{2}})]'\Big|_{\delta=0}$$

$$= -\sum_{j=1}^{k}\mathrm{Tr}\left(\Upsilon\left([D(\delta)^{-\frac{1}{2}}]'\lambda_j^{(u)}\Phi_j^{(u)} + \lambda_j^{(u)}\Phi_j^{(u)}[D(\delta)^{-\frac{1}{2}}]' + [\lambda_j(\delta)]'\Phi_j^{(u)} + \lambda_j^{(u)}[\Phi_j(\delta)]'\right)\right)$$

$$= \sum_{j=1}^{k}\mathrm{Tr}\left(\Upsilon\left(\frac{\lambda_j^{(u)}}{2}D_l\Phi_j^{(u)} + \frac{\lambda_j^{(u)}}{2}\Phi_j^{(u)}D_l - [\lambda_j(\delta)]'\Phi_j^{(u)} - \lambda_j^{(u)}[\Phi_j(\delta)]'\right)\right)$$

$$= \mathcal{M}_a' + \mathcal{M}_b' + \mathcal{M}_c',$$

where

$$\mathcal{M}_a' = \sum_{j=1}^{k}\frac{\lambda_j^{(u)}}{2}\,\mathrm{Tr}\left(\Upsilon\left(D_l\Phi_j^{(u)} + \Phi_j^{(u)}D_l\right)\right),$$

$$\mathcal{M}_b' = -\sum_{j=1}^{k}\mathrm{Tr}\left(\Upsilon[\lambda_j(\delta)]'\Phi_j^{(u)}\right) = -\sum_{j=1}^{k}\mathrm{Tr}\left((\mathbb{1}^\top - \lambda_j^{(u)}D_l)\Phi_j^{(u)}\right)\mathrm{Tr}\left(\Upsilon\Phi_j^{(u)}\right)$$

$$= -\sum_{j=1}^{k}\mathrm{Tr}\left((\mathbb{1}^\top - \lambda_j^{(u)}D_l)\Phi_j^{(u)}\Upsilon\Phi_j^{(u)}\right),$$

$$\mathcal{M}_c' = -\sum_{j=1}^{k}\mathrm{Tr}\left(\Upsilon\lambda_j^{(u)}[\Phi_j(\delta)]'\right)$$

$$= -\sum_{j=1}^{k}\mathrm{Tr}\left(\sum_{i\neq j}^{N}\frac{\lambda_j^{(u)}}{\lambda_j^{(u)} - \lambda_i^{(u)}}(\Upsilon\Phi_i^{(u)}(\mathbb{1}^\top - \frac{\lambda_j^{(u)} + \lambda_i^{(u)}}{2}D_l)\Phi_j^{(u)} + \Upsilon\Phi_j^{(u)}(\mathbb{1}^\top - \frac{\lambda_j^{(u)} + \lambda_i^{(u)}}{2}D_l)\Phi_i^{(u)})\right)$$

$$= -\sum_{j=1}^{k}\mathrm{Tr}\left(\sum_{i\neq j}^{N}\frac{\lambda_j^{(u)}}{\lambda_j^{(u)} - \lambda_i^{(u)}}\left((\Phi_j^{(u)}\Upsilon\Phi_i^{(u)} + \Phi_i^{(u)}\Upsilon\Phi_j^{(u)})(\mathbb{1}^\top - \frac{\lambda_j^{(u)} + \lambda_i^{(u)}}{2}D_l)\right)\right)$$

$$= -\sum_{j=1}^{k}\mathrm{Tr}\left(\sum_{i\neq j, i\leq k}\frac{\lambda_j^{(u)}}{\lambda_j^{(u)} - \lambda_i^{(u)}}\left((\Phi_j^{(u)}\Upsilon\Phi_i^{(u)} + \Phi_i^{(u)}\Upsilon\Phi_j^{(u)})(\mathbb{1}^\top - \frac{\lambda_j^{(u)} + \lambda_i^{(u)}}{2}D_l)\right)\right)$$

$$- \sum_{j=1}^{k}\mathrm{Tr}\left(\sum_{i=k+1}^{N}\frac{\lambda_j^{(u)}}{\lambda_j^{(u)} - \lambda_i^{(u)}}\left((\Phi_j^{(u)}\Upsilon\Phi_i^{(u)} + \Phi_i^{(u)}\Upsilon\Phi_j^{(u)})(\mathbb{1}^\top - \frac{\lambda_j^{(u)} + \lambda_i^{(u)}}{2}D_l)\right)\right)$$

$$= -\sum_{j=1}^{k}\mathrm{Tr}\left(\sum_{i<j}\left(\frac{\lambda_j^{(u)}}{\lambda_j^{(u)} - \lambda_i^{(u)}} + \frac{\lambda_i^{(u)}}{\lambda_i^{(u)} - \lambda_j^{(u)}}\right)\left((\Phi_j^{(u)}\Upsilon\Phi_i^{(u)} + \Phi_i^{(u)}\Upsilon\Phi_j^{(u)})(\mathbb{1}^\top - \frac{\lambda_j^{(u)} + \lambda_i^{(u)}}{2}D_l)\right)\right)$$

$$- \sum_{j=1}^{k}\mathrm{Tr}\left(\sum_{i=k+1}^{N}\frac{\lambda_j^{(u)}}{\lambda_j^{(u)} - \lambda_i^{(u)}}\left((\Phi_j^{(u)}\Upsilon\Phi_i^{(u)} + \Phi_i^{(u)}\Upsilon\Phi_j^{(u)})(\mathbb{1}^\top - \frac{\lambda_j^{(u)} + \lambda_i^{(u)}}{2}D_l)\right)\right)$$

$$= -\sum_{j=1}^{k} \mathrm{Tr}\left( \sum_{i<j} \left( (\Phi_j^{(u)} \Upsilon \Phi_i^{(u)} + \Phi_i^{(u)} \Upsilon \Phi_j^{(u)})(\mathrm{I\!I}^\top - \frac{\lambda_j^{(u)} + \lambda_i^{(u)}}{2} D_l) \right) \right)$$

$$-\sum_{j=1}^{k} \mathrm{Tr}\left( \sum_{i=k+1}^{N} \frac{\lambda_j^{(u)}}{\lambda_j^{(u)} - \lambda_i^{(u)}} \left( (\Phi_j^{(u)} \Upsilon \Phi_i^{(u)} + \Phi_i^{(u)} \Upsilon \Phi_j^{(u)})(\mathrm{I\!I}^\top - \frac{\lambda_j^{(u)} + \lambda_i^{(u)}}{2} D_l) \right) \right)$$

$$= -\sum_{j=1}^{k} \mathrm{Tr}\left( \sum_{i \neq j, i \leq k} \frac{1}{2} \left( (\Phi_j^{(u)} \Upsilon \Phi_i^{(u)} + \Phi_i^{(u)} \Upsilon \Phi_j^{(u)})(\mathrm{I\!I}^\top - \frac{\lambda_j^{(u)} + \lambda_i^{(u)}}{2} D_l) \right) \right)$$

$$-\sum_{j=1}^{k} \mathrm{Tr}\left( \sum_{i=k+1}^{N} \frac{\lambda_j^{(u)}}{\lambda_j^{(u)} - \lambda_i^{(u)}} \left( (\Phi_j^{(u)} \Upsilon \Phi_i^{(u)} + \Phi_i^{(u)} \Upsilon \Phi_j^{(u)})(\mathrm{I\!I}^\top - \frac{\lambda_j^{(u)} + \lambda_i^{(u)}}{2} D_l) \right) \right).$$

Thus, we have:

$$\mathcal{M}_b' + \mathcal{M}_c' = -\sum_{j=1}^{k} \mathrm{Tr}\left( \sum_{i=1}^{k} \frac{1}{2} \left( (\Phi_j^{(u)} \Upsilon \Phi_i^{(u)} + \Phi_i^{(u)} \Upsilon \Phi_j^{(u)})(\mathrm{I\!I}^\top - \frac{\lambda_j^{(u)} + \lambda_i^{(u)}}{2} D_l) \right) \right)$$

$$-\sum_{j=1}^{k} \mathrm{Tr}\left( \sum_{i=k+1}^{N} \frac{\lambda_j^{(u)}}{\lambda_j^{(u)} - \lambda_i^{(u)}} \left( (\Phi_j^{(u)} \Upsilon \Phi_i^{(u)} + \Phi_i^{(u)} \Upsilon \Phi_j^{(u)})(\mathrm{I\!I}^\top - \frac{\lambda_j^{(u)} + \lambda_i^{(u)}}{2} D_l) \right) \right),$$

$$\mathcal{M}_a' = \sum_{j=1}^{k} \frac{\lambda_j^{(u)}}{2} \mathrm{Tr}\left( \Upsilon \left( D_l \Phi_j^{(u)} + \Phi_j^{(u)} D_l \right) \right)$$

$$= \sum_{j=1}^{k} \frac{\lambda_j^{(u)}}{2} \mathrm{Tr}\left( \left( \Phi_j^{(u)} \Upsilon + \Upsilon \Phi_j^{(u)} \right) D_l \right)$$

$$= \sum_{j=1}^{k} \frac{\lambda_j^{(u)}}{2} \mathrm{Tr}\left( \left( \Phi_j^{(u)} \Upsilon \sum_{i=1}^{N} \Phi_i^{(u)} + \sum_{i=1}^{N} \Phi_i^{(u)} \Upsilon \Phi_j^{(u)} \right) D_l \right)$$

$$= \sum_{j=1}^{k} \mathrm{Tr}\left( \sum_{i=1}^{N} \frac{\lambda_j^{(u)}}{2} \left( \Phi_j^{(u)} \Upsilon \Phi_i^{(u)} + \Phi_i^{(u)} \Upsilon \Phi_j^{(u)} \right) D_l \right).$$

Then $\left[ \mathcal{M}_{kms-all}(\delta) \right]' \big|_{\delta=0} / \eta_1$ is given by:

$$\mathcal{M}_a' + \mathcal{M}_b' + \mathcal{M}_c' = -\sum_{j=1}^{k} \mathrm{Tr}\left( \sum_{i=1}^{k} \frac{1}{2} \left( (\Phi_j^{(u)} \Upsilon \Phi_i^{(u)} + \Phi_i^{(u)} \Upsilon \Phi_j^{(u)})(\mathrm{I\!I}^\top - \frac{3\lambda_j^{(u)} + \lambda_i^{(u)}}{2} D_l) \right) \right)$$

$$-\sum_{j=1}^{k} \mathrm{Tr}\left( \sum_{i=k+1}^{N} \frac{\lambda_j^{(u)}}{\lambda_j^{(u)} - \lambda_i^{(u)}} \left( (\Phi_j^{(u)} \Upsilon \Phi_i^{(u)} + \Phi_i^{(u)} \Upsilon \Phi_j^{(u)})(\mathrm{I\!I}^\top - \lambda_j^{(u)} D_l) \right) \right)$$

$$= -\sum_{j=1}^{k} \mathrm{Tr}\left( \sum_{i=1}^{k} \frac{1}{2} \left( (\Phi_j^{(u)} \Upsilon \Phi_i^{(u)} + \Phi_i^{(u)} \Upsilon \Phi_j^{(u)})(\mathrm{I\!I}^\top - 2\lambda_j^{(u)} D_l) \right) \right)$$

$$-\sum_{j=1}^{k} \mathrm{Tr}\left( \sum_{i=k+1}^{N} \frac{\lambda_j^{(u)}}{\lambda_j^{(u)} - \lambda_i^{(u)}} \left( (\Phi_j^{(u)} \Upsilon \Phi_i^{(u)} + \Phi_i^{(u)} \Upsilon \Phi_j^{(u)})(\mathrm{I\!I}^\top - \lambda_j^{(u)} D_l) \right) \right)$$

$$= -\sum_{j=1}^{k} \sum_{i=1}^{k} v_i^{(u)\top} \Upsilon v_j^{(u)} \cdot v_i^{(u)\top} (\mathrm{I\!I}^\top - 2\lambda_j^{(u)} D_l) v_j^{(u)}$$

$$-\sum_{j=1}^{k} \sum_{i=k+1}^{N} \frac{2\lambda_j^{(u)}}{\lambda_j^{(u)} - \lambda_i^{(u)}} v_i^{(u)\top} \Upsilon v_j^{(u)} \cdot v_i^{(u)\top} (\mathrm{I\!I}^\top - \lambda_j^{(u)} D_l) v_j^{(u)}.$$

We can represent $\frac{\lambda_j^{(u)}}{\lambda_j^{(u)}-\lambda_i^{(u)}} = 1 + \sum_{p=1}^{\infty}(\frac{\lambda_i^{(u)}}{\lambda_j^{(u)}})^p$. Denote the residual term as :

$$\mathcal{M}_e' = -\sum_{j=1}^{k}\sum_{i=k+1}^{N}\sum_{p=1}^{\infty}2(\frac{\lambda_i^{(u)}}{\lambda_j^{(u)}})^p v_i^{(u)\top}\Upsilon v_j^{(u)} \cdot v_i^{(u)\top}(\amalg^\top - \lambda_j^{(u)}D_l)v_j^{(u)} = O(\frac{1}{\mathcal{G}_k}).$$

We then have:

$$
\begin{aligned}
&\frac{1}{\eta_1}\big[\mathcal{M}_{kms-all}(\delta)\big]'\Big|_{\delta=0}\\
&= -\operatorname{Tr}(V_k^{(u)\top}\Upsilon V_k^{(u)}\cdot V_k^{(u)\top}\amalg^\top V_k^{(u)}) + 2\operatorname{Tr}(V_k^{(u)\top}\Upsilon V_k^{(u)}\cdot \Sigma_k^{(u)}V_k^{(u)\top}D_l V_k^{(u)})\\
&\quad - 2\operatorname{Tr}(V_\varnothing^{(u)\top}\Upsilon V_k^{(u)}\cdot V_k^{(u)\top}\amalg^\top V_\varnothing^{(u)}) + 2\operatorname{Tr}(V_\varnothing^{(u)\top}\Upsilon V_k^{(u)}\cdot \Sigma_k^{(u)}V_k^{(u)\top}D_l V_\varnothing^{(u)}) + \mathcal{M}_e'\\
&= -\operatorname{Tr}(\Upsilon V_k^{(u)}V_k^{(u)\top}\amalg^\top V_k^{(u)}V_k^{(u)\top}) + 2\operatorname{Tr}(\Upsilon \tilde{A}_k^{(u)}D_l V_k^{(u)}V_k^{(u)\top})\\
&\quad - 2\operatorname{Tr}(\Upsilon V_k^{(u)}V_k^{(u)\top}\amalg^\top(I_N - V_k^{(u)}V_k^{(u)\top})) + 2\operatorname{Tr}(\Upsilon \tilde{A}_k^{(u)}D_l(I_N - V_k^{(u)}V_k^{(u)\top})) + \mathcal{M}_e'\\
&= -2\operatorname{Tr}(\Upsilon V_k^{(u)}V_k^{(u)\top}\amalg^\top) + 2\operatorname{Tr}(\Upsilon \tilde{A}_k^{(u)}D_l) + \operatorname{Tr}(\Upsilon V_k^{(u)}V_k^{(u)\top}\amalg^\top V_k^{(u)}V_k^{(u)\top}) + \mathcal{M}_e'\\
&= -2\operatorname{Tr}\left(\Upsilon\left(V_k^{(u)}V_k^{(u)\top}\amalg^\top - \tilde{A}_k^{(u)}D_l - \frac{1}{2}V_k^{(u)}V_k^{(u)\top}\amalg^\top V_k^{(u)}V_k^{(u)\top}\right)\right) + \mathcal{M}_e'\\
&= -\operatorname{Tr}\left(\Upsilon\left(V_k^{(u)}V_k^{(u)\top}\amalg^\top - 2\tilde{A}_k^{(u)}D_l + V_k^{(u)}V_k^{(u)\top}\amalg^\top V_\varnothing V_\varnothing^\top\right)\right) + O(\frac{1}{\mathcal{G}_k}).
\end{aligned}
$$

$\square$

# D    Analysis on Other Contrastive Losses

In this section, we discuss the extension of our graphic-theoretic analysis to one of the most common contrastive loss functions – SimCLR [11]. SimCLR loss is an extended version of InfoNCE loss [68] that achieves great empirical success and inspires a proliferation of follow-up works [5, 8, 12, 26, 34, 69, 77]. Specifically, SupCon [34] extends SimCLR to the supervised setting. GCD [69] and OpenCon [63] further leverage the SupCon and SimCLR losses, and are tailored to the open-world representation learning setting considering both labeled and unlabeled data.

At a high level, we consider a general form of the SimCLR and its extensions (including SupCon, GCD, OpenCon) as:

$$\mathcal{L}_{\text{gnl}}(f; \mathcal{P}_+) = -\frac{1}{\tau} \underset{(x,x^+)\sim\mathcal{P}_+}{\mathbb{E}} \left[ f(x)^\top f(x^+) \right] \quad + \underset{x\sim\mathcal{P}}{\mathbb{E}} \left[ \log \left( \underset{\substack{x'\sim\mathcal{P}\\x\neq x'}}{\mathbb{E}} e^{f(x')^\top f(x)/\tau} \right) \right], \quad (15)$$

where we let the $\mathcal{P}_+$ as the distribution of **positive pairs** defined in Section 3.1. In SimCLR [11], the positive pairs are purely sampled in the *unlabeled case (u)* while SupCon [34] considers the *labeled case (l)*. With both labeled and unlabeled data, GCD [69] and OpenCon [63] sample positive pairs in both cases.

In this section, we investigate an alternative form that eases the theoretical analysis (also applied in [72]):

$$\widehat{\mathcal{L}}_{\text{gnl}}(f; \mathcal{P}_+) = -\frac{1}{\tau} \underset{(x,x^+)\sim\mathcal{P}_+}{\mathbb{E}} \left[ f(x)^\top f(x^+) \right] \quad + \log \left( \underset{\substack{x,x'\sim\mathcal{P}\\x\neq x'}}{\mathbb{E}} e^{f(x')^\top f(x)/\tau} \right) \quad (16)$$

$$\geq \mathcal{L}_{\text{gnl}}(f; \mathcal{P}_+), \quad (17)$$

which serves an upper bound of $\mathcal{L}_{\text{gnl}}(f)$ according to Jensen's Inequality.

**A graph-theoretic view.** Recall in Section 3.1, we define the graph $G(\mathcal{X}, w)$ with vertex set $\mathcal{X}$ and edge weights $w$. Each entry of adjacency matrix $A$ is given by $w_{xx'}$, which denotes the marginal probability of generating the pair for any two augmented data $x, x' \in \mathcal{X}$:

$$w_{xx'} = \eta_u w_{xx'}^{(u)} + \eta_l w_{xx'}^{(l)},$$

and $w_x$ measures the degree of node $x$:

$$w_x = \sum_{x'} w_{xx'}.$$

One can view the difference between SimCLR and its variants in the following way: (1) SimCLR [11] corresponds to $\eta_l = 0$ when there is no labeled case; (2) SupCon [34] corresponds to $\eta_u = 0$ when only labeled case is considered. (3) GCD [69] and OpenCon [63] correspond to the cases when $\eta_u, \eta_l$ are both non-zero due to the availability of both labeled and unlabeled data.

With the define marginal probability of sampling positive pairs $w_{xx'}$ and the marginal probability of sampling a single sample $w_x$, we have:

$$\widehat{\mathcal{L}}_{\text{gnl}}(Z; G(\mathcal{X}, w)) = -\frac{1}{\tau} \sum_{x,x'\in\mathcal{X}} w_{xx'} f(x)^\top f(x') + \log \left( \sum_{\substack{x,x'\in\mathcal{X}\\x\neq x'}} w_x w_{x'} e^{f(x')^\top f(x)/\tau} \right)$$

$$= -\frac{1}{\tau} \text{Tr}(Z^\top A Z) + \log \text{Tr}\left( (D\mathbf{1}_N \mathbf{1}_N^\top D - D^2) \exp(\frac{1}{\tau} Z Z^T) \right).$$

When $\tau$ is large:

$$\widehat{\mathcal{L}}_{\text{simclr}}(Z; G(\mathcal{X}, w)) \approx -\frac{1}{\tau} \text{Tr}(Z^\top A Z) + \log \text{Tr}\left((D\mathbf{1}_N \mathbf{1}_N^\top D - D^2)(\mathbf{1}_N \mathbf{1}_N^\top + \frac{1}{\tau} Z Z^T)\right)$$

$$= -\frac{1}{\tau} \text{Tr}(Z^\top A Z) + \log(1 + \frac{\frac{1}{\tau} \text{Tr}(Z^\top (D\mathbf{1}_N \mathbf{1}_N^\top D - D^2)Z)}{\text{Tr}(D)^2 - \text{Tr}(D^2)}) + \text{const}$$

$$\approx -\frac{1}{\tau} \text{Tr}(Z^\top A Z) + \frac{\frac{1}{\tau} \text{Tr}(Z^\top (D\mathbf{1}_N \mathbf{1}_N^\top D - D^2)Z)}{\text{Tr}(D)^2 - \text{Tr}(D^2)} + \text{const}$$

$$= -\frac{1}{\tau} \text{Tr}\left(Z^\top (A - \frac{D\mathbf{1}_N \mathbf{1}_N^\top D - D^2}{\text{Tr}(D)^2 - \text{Tr}(D^2)})Z\right) + \text{const}.$$

If we further consider the constraint that the $Z^\top Z = I$, minimizing $\widehat{\mathcal{L}}_{\text{simclr}}(Z; G(\mathcal{X}, w))$ boils down to the eigenvalue problem such that $Z$ is formed by the top-$k$ eigenvectors of matrix $(A - \frac{D\mathbf{1}_N \mathbf{1}_N^\top D - D^2}{\text{Tr}(D)^2 - \text{Tr}(D^2)})$. Recall that our main analysis for Theorem 4.2 and Theorem 4.3 is based on the insight that the feature space is formed by the top-$k$ eigenvectors of the normalized adjacency matrix $D^{-\frac{1}{2}} A D^{-\frac{1}{2}}$. Viewed in this light, the same analysis could be applied to the SimCLR loss as well, which only differs in the concrete matrix form. We do not include the details in this paper but leave it as a future work.

# E  Additional Experiments Details

## E.1  Experimental Details of Toy Example

**Recap of set up**. In Section 4.1 we consider a toy example that helps illustrate the core idea of our theoretical findings. Specifically, the example aims to cluster 3D objects of different colors and shapes, generated by a 3D rendering software [31] with user-defined properties including colors, shape, size, position, etc. Suppose the training samples come from three shapes, $\mathcal{X}_{\square}, \mathcal{X}_{\bigcirc}, \mathcal{X}_{\ominus}$. Let $\mathcal{X}_{\square}$ be the sample space with **known** class, and $\mathcal{X}_{\bigcirc}, \mathcal{X}_{\ominus}$ be the sample space with **novel** classes. Further, the two novel classes are constructed to have different relationships with the known class. Specifically, the toy dataset contains elements with 5 unique types:

$$\mathcal{X} = \mathcal{X}_{\square} \cup \mathcal{X}_{\bigcirc} \cup \mathcal{X}_{\ominus},$$

where

$$\mathcal{X}_{\square} = \{x_{\square}, x_{\square}\},$$
$$\mathcal{X}_{\bigcirc} = \{x_{\bigcirc}, x_{\bigcirc}\},$$
$$\mathcal{X}_{\ominus} = \{x_{\ominus}\}.$$

**Experimental details for Figure 3(b)**. We rendered 2500 samples for each type of data. In total, we have 12500 samples. For known class $\mathcal{X}_{\square}$, we randomly select $50\%$ as labeled data and treat the rest as unlabeled. For training, we use the same data augmentation strategy as in SimSiam [12]. We use ResNet18 and train the model for 40 epochs (sufficient for convergence) with a fixed learning rate of 0.005, using SORL defined in Eq. (6). We set $\eta_l = 0.04$ and $\eta_u = 1$, respectively. Our visualization is by PyTorch implementation of UMAP [43], with parameters (`n_neighbors=30, min_dist=1.5, spread=2, metric=euclidean`).

## E.2  Experimental Details for Benchmarks

**Hardware and software.**  We run all experiments with Python 3.7 and PyTorch 1.7.1, using NVIDIA GeForce RTX 2080Ti and A6000 GPUs.

**Training settings.**  For a fair comparison, we use ResNet-18 [25] as the backbone for all methods. Similar to [7], we pre-train the backbone using the unsupervised Spectral Contrastive Learning [23] for 1200 epochs. The configuration for the pre-training stage is consistent with [23]. Note that the pre-training stage does not incorporate any label information. At the training stage, we follow the same practice in [7, 63], and train our model $f(\cdot)$ by only updating the parameters of the last block of ResNet. In addition, we add a trainable two-layer MLP projection head that projects the feature from the penultimate layer to an embedding space $\mathbb{R}^k$ ($k = 1000$). We use the same data augmentation strategies as SimSiam [12, 23]. For CIFAR-10, we set $\eta_l = 0.25, \eta_u = 1$ with training epoch 100, and we evaluate using features extracted from the layer preceding the projection. For CIFAR-100, we set $\eta_l = 0.0225, \eta_u = 3$ with 400 training epochs and assess based on the projection layer's features. We use SGD with momentum 0.9 as an optimizer with cosine annealing (lr=0.05), weight decay 5e-4, and batch size 512.

**Evaluation settings.**  At the inference stage, we evaluate the performance in a transductive manner (evaluate on $\mathcal{D}_u$). We run a semi-supervised K-means algorithm as proposed in [69]. We follow the evaluation strategy in [7] and report the following metrics: (1) classification accuracy on known classes, (2) clustering accuracy on the novel data, and (3) overall accuracy on all classes. The accuracy of the novel classes is measured by solving an optimal assignment problem using the Hungarian algorithm [36]. When reporting accuracy on all classes, we solve optimal assignments using both known and novel classes.

