# OpenReview forum: "A Graph-Theoretic Framework for Understanding Open-World Semi-Supervised Learning"
_NeurIPS.cc/2023/Conference — NeurIPS 2023 spotlight_

### Official Review · Reviewer_1J6p · 2023-07-04

**Soundness:** 3 good
**Presentation:** 3 good
**Contribution:** 4 excellent
**Rating:** 6
**Confidence:** 3

**Summary:**

The authors describe a theoretical framework for contrastive learning of representations in an open-world setting, where both labelled data and unlabelled data of potentially new classes is available. They explicitly describe the graph encoding positive sample connections and formulate a contrastive loss that amounts to a spectral decomposition of the adjacency matrix. Using this framework, they prove two theorems about the change in clustering quality of their representations via k-means when incorporating some labels. The main take-away is that labelled data helps to cluster another class if this other class has strong connections to the labelled class via data augmentation and was previously poorly clustered. Their method also outperforms several competitors for open-world and self-supervised representation learning with subsequent clustering.


**Strengths:**

I appreciate the theoretical nature of the paper. Understanding the behavior of machine learning models when a clear learning signal via labels is unavailable or just partially available is important for their development and trustworthiness. It is also encouraging that their framework does not only allow a formal investigation but also holds up practically. I find the paper is well written with an appropriate deferral of more technical parts to the appendix. The authors also discuss connections of their work to the more common contrastive SimCLR loss. Their theory seems to be novel for the setting of open-world representation learning.

**Weaknesses:**

**Strength of different types of augmentations:**
In the illustrative example in section 4.1. it is assumed that the probability $\tau_c$ of augmenting across classes if there is a shared attribute (e.g. augmenting a blue cube to a blue ball) is larger than the probability $\tau_s$ of just changing attributes within a class (blue ball to red ball). I find this quite counterintuitive.
First, I am surprised that $\tau_c$ should be non-zero. This might be achievable in the toy setting, but for any real world data, say CIFAR10, it is very unlikely, if not impossible, to augment two images of different classes to the same image. Maybe a plausible change of class can be achieved via cropping (cropping an image of a dog in front of flowers to just the flower) but it still seems unlikely that there should be another image in the dataset that augments to the same view. The same argument applies, to a lesser extend, also to the question if two images of the same class can be augmented to the same view (positivity of $\tau_s$). In the toy setting, I also wonder how positions and sizes were treated. Can a large blue ball in the top left corner only augment to a large blue cube in the top left corner or to any blue cube?
Second, I am surprised by the order of augmentation probabilities. Is it not more likely for two data points of the same class to augment to the same view than for two data points of different classes? A similar requirement seems to relevant for Theorem 4.2 (l255 ff). How crucial are these assumptions (or corresponding ones) for the outcome of the illustrative example, the main theory, and the empirical validation? How relevant does this make the theoretical analysis in real settings?

**Empirical illustration of the main theorems:**
It would strengthen the paper if the authors had included experiments that showed how the predicted quantities in Thm 4.1 and 4.2 behave empirically in real / toy settings. In particular, a graph that shows the $\Delta(\delta)$ terms in both theorems for different $\delta$ values both according to their theory and in terms of actual experiments. This would also make it more evident how restrictive the assumptions of the theorems are and how tight the inequality is. The size and evolution of the three summands in the last equation of Thm 4.2 would also be interesting.


**Minor:**
- I think that highlighting which feature is more separable in Fig 3b) is a bit problematic. First, UMAP plots are known to artificially tear apart structures. Second, the shape separability in 3b1) is also very good (but admittedly less obvious when looking at the plot as the color of the marker is much more prominent that its shape in the 2D plot). Therefore, I would recommend to rather stress the fact that when incorporating labels, the islands of the same shape but of different color seem to be much closer together (i.e. much less separable by color) than without labels.
- It would have helped my understanding of the paper if it was stated more explicitly that label information is made available by providing the label of existing but previously unlabelled points as opposed to adding new labelled points.
- I would suggest to also assign the bottom matrix in Fig 2b) as $\eta_l A^{(l)}$ as in the line above and then add another line $A=...$ below.

**Questions:**

**Size of the data augmentation graph:**
The node set of the graph described in 3.1 is the set of all augmented views of all available data. Why is this even finite? Consider, for instance, image data. Are there not infinitely many ways to color jitter even a single image? Or is the idea to only consider a fixed and finite set of augmentations of each data point that is selected prior to training? This would be different than what is typically done in contrastive learning, where augmentations are applied on the fly. Even when considering a fixed set of augmentations, the size of the graph would have to be many times the size of the original dataset, right?

**Empirical Validation:**
- Were the same positive samples (including label information) used when comparing to SimCLR? From the discussion in appendix D, I think they were not. If that is the case and the SimCLR experiment is purely self-supervised, why is there such a large difference between the known, novel and all classes? Shouldn't all the completely un-/self-supervised methods be agnostic to which class has labels (which they cannot use anyway)?
- Using a ResNet-18 typically leads to about 90% linear probing and kNN accuracy on CIFAR10 for SimCLR. I appreciate that the evaluation is different and k-means is used to infer the clusters here. Nevertheless, I am surprised by the drastic drop in performance. Do the authors have an idea why this happens? Do the same SimCLR representations still achieve about 90% accuracy with linear probing / kNN classifier?
- I appreciate that the paper is about the open-world setting, where there might be novel classes without any labels, so that one can only do clustering and not classification for evaluation. Still the label information is available for CIFAR10 and CIFAR100 and I would have found it interesting to see how the proposed method fares under the SimCLR evaluation protocol with a linear probe / kNN classifier. This setting is standard in the self-supervised learning world and thus makes the quality of the proposed method more interpretable to an audience more familiar with self-supervised rather than open-world representation learning. I assume SORL should outperform SimCLR since it has some label information available at train time, right?
- Why is the performance for the "known" classes lower than that of the "novel" classes and why is the performance for "all" classes (typically) worse still than for the "novel" classes for the methods above the horizontal line in Table 1? For the open-world methods below the horizontal line, the performance is (typically) best on the "known" class, worse on the "novel" class and somewhere in between for "all" classes. This it what I would have intuitively expected.
- The method LSD-C [1] achieves over 80% clustering accuracy on CIFAR-10 without using any labels (and no supervised training of a shallow classifier). It might therefore be a relevant competitor.

**Minor:**
- The batch size of 512 is rather small for contrastive learning. Why is that?
- Are the different islands of points that share both color in shape in Fig 3b meaningful? Do they encode perhaps the size or position?

**References:**
[1] Rebuffi, Sylvestre-Alvise, et al. "Lsd-c: Linearly separable deep clusters." Proceedings of the IEEE/CVF international conference on computer vision. 2021.


**Limitations:**

Limitations were not explicitly discussed. I suggest a more detailed discussion of how their theory and assumptions hold up on real world data, see weaknesses above.

---

> ### Author Rebuttal · Authors · 2023-08-08
>
>
> We thank the reviewer for the insightful questions! Below we address each of your comments in detail.
>
> #### **- Discussion on the augmentation graphs.**
> We have noted the reviewers' concerns regarding the definition of the augmentation graph from several angles, and we address these concerns as follows:
>
>  1. *The feasibility of augmenting two real images to the same view.*
> We agree that most images cannot be augmented from images with dissimilar content when limiting the augmentation strategies to commonly applied techniques like cropping and color jittering. However, it becomes achievable once we expand the strategy to include the use of generative models. Though it's not necessary to apply generative augmentation in contrastive learning, the research community (Haochen et al., 2021; 2022; Shen et al., 2022) is actively utilizing the concept of the augmentation graph to theoretically explain the practical success of contrastive learning.
> 2. *The finiteness of the augmentation graph.*
> Though $N$ can be incredibly large in practice, it remains finite. It is upper-bounded by $256^{3 \times W \times H}$, considering a $W \times H$ RGB image. Despite its finiteness, the graph remains intractable in practice when considering all augmentation views. This complexity necessitates the use of SORL to learn a low-rank representation space, enabling us to approximate the graph rather than directly optimizing $\mathcal{L}_{mf}$.
> 3. *The order of the augmentation probability.*
> In the main theory sections (4.1 & 4.2), we do not make any assumptions regarding the order of augmentation probability since it is inherently determined by the augmentation strategies.
>
> #### **- Justification on the graph setup for the toy example.**
>
> We are happy to provide justification for the reviewer’s concern about the setup in the toy example from several aspects:
>
> 1. *Why only consider shape and color?*.
> We fully agree with the reviewer's insight that real 3D-shape images include a broad range of properties, including color, shape, angle, position, size, material, etc. However, our primary objective with the toy example is to elucidate the main intuition. Therefore, we have chosen to focus solely on the two most salient features, color and shape, in constructing our toy augmentation graph.
> 2. *Why assume a non-zero $\tau_c$?*.
> Given our simplification of the property set to {color, shape}, we also presume the existence of augmentation strategies capable of permuting colors and shapes. As a result, $\tau_c$  and $\tau_s$ are two non-zero values in our toy example.
> 3. *Why assume $\tau_c$ >  $\tau_s$?*
> It's essential to note that this assumption only serves to facilitate our illustration and does not impact the main theorem or empirical validation. Our principal goal here is to demonstrate that the addition of labels can alter clustering outcomes. In Figure 3(a), we depict how adding labels to cubes can reverse the incorrect trend where samples are clustered by color. Similarly, one could also construct a toy example where $\tau_s$ > $\tau_c$, demonstrating that adding labels to "red samples" can invert the clustering results by shapes.
>
>
>
> #### **- Empirical illustration of the main theorems.**
>
> Great suggestions! In response, we are pleased to introduce numerical verification to affirm the correctness of Theorems 4.1 and 4.2 with our toy example. **Due to the space limitation, we provide the details in the [additional rebuttal pdf](https://openreview.net/attachment?id=09O88IQijF&name=pdf)**.
>
>
> It is important to note that applying Theorems 4.1 and 4.2 necessitates explicit knowledge of the adjacency matrix. However, obtaining such a matrix becomes intractable in real datasets, particularly when considering all augmentation views of source images.
>
> #### **- Justification on results of SimCLR.**
>
> We are happy to collaborate with the reviewer to justify the baseline results, which are directly cited from the prior work ORCA (Cao et al., 2021). To reproduce the results, we train SimCLR on CIFAR-10, using the code available at https://github.com/HobbitLong/SupContrast. Our training achieved an All/Novel/Seen Accuracy of 61.1/81.0/72.1, which is notably higher than what was reported in ORCA. Furthermore, we noted that K-means results tend to fluctuate when accuracy is low and class number is small, as with CIFAR-10. An unfortunate initialization of class centers can significantly downgrade clustering performance. This observation might provide insights into why the magnitude order can sometimes diverge from expectations.
>
>
>
>
> #### **- Other comments and questions.**
>
> + *Measuring SORL/SimCLR on Linear Probing/KNN classifier protocol.* As suggested, we provide the comparison of SimCLR and SORL below. By having more label information, SORL performs stronger than SimCLR on CIFAR-10, which follows the intuition.
>
> | Method | Linear Probing | KNN classifier|
> |--|--|--|
> | SimCLR | 87.6 | 86.2 |
> | SORL | 92.9 | 91.4 |
> + *Performance on LSD-C*. For the reviewer's interest, the All/Novel/Seen accuracy on CIFAR-10 is 82.1/88.8/87.6 respectively.
> + *Why do we use 512 as a batch size?* To ensure a fair comparison in training setup, we follow the protocol in existing works including ORCA (Cao et al., 2021) and OpenCon (Sun et al., 2023).
> + *What do different islands of points mean in Figure 3(b)?* Great catch! When we generate these 3D shapes, there are other variants like materials (rubber, metal) and sizes (big, small). Therefore, the sub-clusters are very likely to represent other variants.
>
>
> #### **- Writing suggestions.**
> + *The conclusion of UMAP visualization.* Thanks for the suggestions! We will revise the claim as suggested in Figure 3(b).
> + *The explicitness of the labeling perturbation setup.* Great comment! See changes in the global response.
> + *Adjustment in the layout of Figure 2.* Fixed, thanks for the suggestion!
> + *Add the limitation discussion.* Absolutely! See global response.

---

> > ### Comment · Reviewer_1J6p · 2023-08-11
> >
> > Thank you for your reply!
> > **Discussion of the augmentation graph:**
> > Your clarifications are helpful, especially the mention of generative augmentations and the limited relevance of the order of augmentation probabilities for the Theory in Sec 4.
> >
> > **Justification on the graph setup for the toy example:**
> > - *Why only consider shape and color* Of course, a toy example deliberately has fewer properties than a real example. My confusion stemmed from a misunderstanding of the augmentations. I had not realised that all the augmentation probabilities are simply completely independent of the position and size of the shape, e.g., an object has the same probability to be augmented to any other position or size without changing shape and color, namely $\tau_1$. This should actually have been sufficiently clear from Fig 2a). So in a way, there is "perfect augmentation" to unlearn the irrelevance of size and position.
> > - *Why assume a non-zero $\tau_c$?* I now understand that the order of the augmentation probabilies does not matter for correctness of the theory in Section 4. But the connection of unlabelled to labelled data, i.e., $\mathfrak{l}$ does enter Thm 4.1 and Thm 4.2. In that light, I appreciate the first part of the proposed limitation statement in the general comment.
> > - *Why assume $\tau_c > \tau_s$?* Thanks for clarifying that the validity of the theory in Sec 4 does not hinge on the order of these probabilities. Do I understand correctly that choosing $\tau_c > \tau_s$ is a deliberately problematic setting, in which augmentations maintaining the category of interest (shapes) are less likely than augmentations that change it and instead maintain a nuisance property (color)? Without labels this augmentation setting would naturally cluster by color rather by shape, as depicted in Fig 3a) Adding the labels for shape cube rectifies the erroneous focus of the augmentations on the color. If one instead chose $\tau_c < \tau_s$, no fix would be required because the resulting clustering is automatically by shape (although perhaps there is still a marginal benefit from using labels for some shape). I think I had missed the first part of this, i.e., that there is anything to fix because the augmentations strengths do not align with the category of interest. From what I understand, a situation similar to $\tau_c > \tau_s$ is also the one in which Thm 4.2 offers the largest benefit: high connection to the labelled class and high inter class similarity (from large $\tau_c$) and low intra-class similarity (from low $\tau_s$). Thanks again for the clarification!
> >
> > **Empirical illustration of the main theorems:**
> > Thank you for including these results in the additional pdf! Especially the main term in Thm 4.1. is impressively accurate. Thanks for clarifying in the proposed limitations section that one needs access to the full adjacency matrix to evaluate the terms in Thms 4.1, 4.2. Sorry if this is a trivial question, but I do not get the connection entry in Tab 2. From the top of page 13, I thought it would be $\mathfrak{l}_{sphere} - 1 / N = 0.5\tau_c -0.0001 = 0.1499$. What am I missing?
> >
> > **Justification on results of SimCLR:**
> > Thank you for computing linear probing and kNN classifier accurarcies on your SimCLR representations. This helps dissecting which part of the performance drop is due to the evaluation protocol and which is due to the representations themselves. It seems that, indeed, the main difference in numbers is due to the evaluation setting. Nevertheless, the linear probing and kNN classifier accuracies are still lower than what as been reported for a ResNet18 on CIFAR10, e.g., in https://openreview.net/pdf?id=B8a1FcY0vi, Tab. 1. It will probably be difficult and not curcial to pinpoint exactly where the difference comes from. Just to confirm my understanding: The SimCLR representation learning is completely unsupervised and uses all of CIFAR10 (train+test set) during this unsupervised training, right?
> >
> > **Performance on LSD-C:**
> > Many thanks for evaluating this additional method. I find it intruiging that it performs on par with RankStats, despite begin entirely unsupervised.

---

> > > ### Author Response · Authors · 2023-08-15
> > > **Thanks for your response!**
> > >
> > >
> > > Thank you for your invaluable feedback! We're pleased to note that our clarification has resonated, and your understanding of the augmentation graph and toy example aligns perfectly with our intent.
> > >
> > > Addressing your further queries:
> > >
> > > + *On the setting of $\tau_c > \tau_s$:* Yes! We choose the setting deliberately so that the clustering will be misled by the colors. It subsequently offers the space to improve by clustering samples on shapes, upon incorporating labels. Your interpretation from the view of Thm 4.2 also perfectly aligns with the intuition!
> > > + *On the value of $\mathfrak{l}_{sphere}$*: Good catch! At a high level, the discrepancy originates from the scaling factor. To elucidate, our theory hinges on the premise that T's row sums to 1. Given $1 = \tau_1 >> \tau_c > \tau_s$ (footnote on page 5), we can follow the equation in the Appendix (top of page 13), which is simplified and has no normalization. We can fix the glitches by setting $\tau_1 = 0.6$, $\tau_c = 0.3$, $\tau_s=0.1$, the value of $\mathfrak{l}\_{sphere}$ now aligns with the reviewer's proposition of 0.15. In this case, concerning the estimation gap, both $\Delta_{kms}(\delta)$ and the estimation now become 0.0027 (when $\delta=0.01$). For the broader audience, we will make the simulation code open-sourced together with the SORL implementation upon the paper's publication.
> > > + *On SimCLR's training setup*: The discrepancy in the SimCLR's performance requires a deeper examination of the different training settings. For the reviewer's reference, we directly use the training script at https://github.com/HobbitLong/SupContrast. Regarding the question, the answer is **yes** for the purely unsupervised setting but **no** for the `train+test` split. Specifically, we perform the unsupervised training on the `train` split but report KNN/LP performance on the `test` split.
> > >
> > > Thank you again for the comments and suggestions to help us improve the manuscript!

---

> > > > ### Comment · Reviewer_1J6p · 2023-08-15
> > > >
> > > > Thank you for the additional information! I maintain my score.

---

### Official Review · Reviewer_66eW · 2023-07-04

**Soundness:** 4 excellent
**Presentation:** 4 excellent
**Contribution:** 3 good
**Rating:** 7
**Confidence:** 3

**Summary:**

This paper presents spectral open-world representation learning that aims to learn low-rank approximation of a constructed adjacency matrix. From this perspective, the authors study how the label information help (or hurt) the classification performance by analyzing the error bound. Experiments on image classification benchmarks show the effectiveness of the proposed approach.

**Strengths:**

1. The formulation of open-world representation learning problem as a low-rank approximation of a weighted graph adjacency matrix is novel and reasonable.
2. The representation is clear. The provided illustrative example is helpful for understanding the theory.
3. The new ORL approach itself is simple yet effective.

**Weaknesses:**

1. The theoretical analysis presented in the main text is only applicable to the proposed SORL approach. Generalizing to other approaches needs independent considerations as stated by the authors in section 6.
2. The empirical validation seems somewhat disconnected with the main theoretical result (theorem 4.1 and 4.2) as it only focuses on the performance of SORL. It would be better to conduct experiments to directly validate the main theorems e.g. by showing the connection between performance improvement by adding different label information and factors affecting $\Delta_{\pi_s}(\delta)$.

**Questions:**

1. Is it possible to generalize the analysis to the SSL and supervised learning settings?
2. Is the spectral gap in theorem 4.1 crucial for the result?

**Limitations:**

Please see the weakness section.

---

> ### Author Rebuttal · Authors · 2023-08-08
>
>
> We thank the reviewer for the insightful questions! Below we address each of your comments in detail.
>
> #### **- Generalization of the theoretical analysis.**
>
> Fair concern! We are happy to expand our thoughts on the generalization of our theoretical analysis within a broader context. The primary objective of introducing SORL is to establish the first theoretical framework that has been lacking in the field of open-world representation learning for several years. We believe that this framework will open up new avenues for a deeper understanding of the effects of labeling perturbation within the community. Specifically:
>
> + *SORL answers the fundamental question common to all methods in open-world representation learning*.
> At a high level, SORL analyzes how the added label changes the representation space that leads to different clustering outcomes. This finding can be generalizable to other methods which may differ in the way of incorporating new labels.
>
> + *SORL paves the way to formally understand open-world representation learning methods.*
> We have laid the groundwork for understanding the closed-form solutions of these learning algorithms from a theoretical standpoint. Viewed from this perspective, the same analytical strategy employed in this paper remains applicable. For instance, we demonstrated in Section 6 that "SimCLR+SupCon" is a factorization of another regularized form of the adjacency matrix, where the perturbation analysis can be equally applied.
>
>  + *SORL is also appealing for practical usage*.
> Our framework is also empirically appealing to use since it can achieve similar or better performance than existing methods on benchmark vision datasets. In particular, on CIFAR-100, we improve upon the best baseline OpenCon (Sun et al., 2022) by 3.4% in terms of overall accuracy.
>
> #### **- Experiments validating theoretical results.**
> Thank you for the excellent recommendations! To delve into the performance disparities among cases with different types of label information, we've structured the experimental setup described below.
>
> We've classified the settings into three categories: `unsup`, `sup-weak`, and `sup-strong`.
>
> + The `unsup` category represents cases without any label information.
> + The `sup-weak` setting involves adding labels to the 50 sub-classes within the first 10 super-classes.
> + The `sup-strong` category entails the distribution of labels evenly across 50 sub-classes spanning all super-classes.
>
> For clarity, we illustrate the configurations for sup-strong and sup-weak in the tables below, showcasing 20 CIFAR-100 classes within four super-classes. Within this configuration, the names of the known classes (those receiving labels) are highlighted in **bold**.
>
> This design ensures that the `sup-strong` setting fosters a more significant relationship between known classes with label information and novel classes compared to the `sup-weak` setting. The empirical results presented in the table below substantiate this insight. The clustering accuracy for the novel classes is ranked as: `sup-strong` > `sup-weak` > `unsup`.
>
>
> + **Empirical comparison**
>
> | Setting | Method |	All | Novel | Seen |
> |--------|--------|----------|-------|-------|
> | `unsup` | SORL | 37.3 | 40.2 | 38.3 |
> | `sup-weak` | SORL | 49.8 | 46.4 | 66.0 |
> | `sup-strong` | SORL | 56.3 | 53.5 | 67.1 |
>
> + **Setting `sup-weak`:**
>
> | Super-class    | Sub-class  |
> |---------------|----------|
> |aquatic mammals | **beaver, dolphin, otter, seal, whale**|
> |fish | **aquarium fish, flatfish, ray, shark, trout**|
> | ... |... |
> |vehicles 1   	| bicycle, bus, motorcycle, pickup truck, train|
> |vehicles 2   	| lawn-mower, rocket, streetcar, tank, tractor|
>
> + **Setting `sup-strong`:**
>
> | Super-class    | Sub-class  |
> |---------------|----------|
> |aquatic mammals | **beaver, dolphin, otter**, seal, whale|
> |fish | **aquarium fish, flatfish**, ray, shark, trout|
> | ... |... |
> |vehicles 1   	| **bicycle, bus, motorcycle**, pickup truck, train|
> |vehicles 2   	| **lawn-mower, rocket**, streetcar, tank, tractor|
>
>
>
> #### **- Generalization to the self-supervised or supervised learning setting.**
> Our theorem explores the effects of labeling perturbation by comparing the performance between the self-supervised learning setting and the setting with partially available label information. This analysis can be further generalized to the supervised learning setting. It can be regarded as a special case within the open-world learning framework, where all classes are known and every sample is labeled.
>
>
> #### **- The role of the spectral gap.**
> Our response is in a two-fold manner.
> + For the clarity of the theoretical results, assuming a large spectral gap is a standard practice in spectral analysis (Joseph et al., 2016; Shen et al., 2022), a convention we also have adhered to in our study.
> + At a high level, it should be noted that the term remains small in most practical scenarios even in the absence of the spectral gap term.
> In more detail, the residual term related to the spectral gap, is included in lines 497-498 of the Appendix. For this term to become large, several conditions must be satisfied simultaneously:  1). spectral gap between $\sigma_k$ and $\sigma_{n}$ must be small. 2). eigenvector $v_{n}$ (corresponding to $\sigma_{n}$) in the null space must encode the critical class information that is missing in the feature space; 3). eigenvector $v_{n}$ must align with the vector $\mathfrak{l}$ which encodes the connection to label data. Although it's theoretically possible to construct a scenario where this occurs, we have chosen not to delve further into this in order to focus on conveying the main intuition of our work.

---

> > ### Comment · Reviewer_66eW · 2023-08-18
> >
> > Thank you for your response. Most of my concerns are addressed. I particularly feel empirical validation of the theory (beyond evaluation of the proposed method) is valuable and encourage the authors to consider incorporating relevant discussions in the paper. I increase soundness to 4.

---

### Official Review · Reviewer_7sHp · 2023-07-06

**Soundness:** 3 good
**Presentation:** 3 good
**Contribution:** 3 good
**Rating:** 7
**Confidence:** 3

**Summary:**

This paper formulate the open-world representation learning using the graph.
This paper provides theoretical analyses for the framework (Thm 3.1) as well as the framework's performance (Thm 4.1 & 4.2).
The experimental results show that the framework outperforms the exiting method in the open-world learning setting.

**Strengths:**

Overall, I enjoy reading this paper and have a positive feeling.

- Formulation. Of Eq. (6). By Thm. 3.1 this formulation corresponds to the SVD of normalized adjacency matrix Eq. (4). See more on the question (*).

- The theoretical insights (Thm 4.1 & 4.2). The analyses using $k$-means measure strengthen the formulation. Particularly, the insight in L251-L258 is interesting; knowing more about green light increases the knowledge about the green apple, but not the flowers. This corresponds to our intuition.

**Weaknesses:**

There's little I can say, but if I need to raise one, I would like to point out on the experiment.

The experiment is on the only one split, but I am curious the other splits. Ideally it'd be nice to have results like Fig 4 in [79], y-axis accuracy vs x-axis # of the known labels.
By seeing this, we can know "how many pictures of the green light do we need to learn in order to know the green apple?"

I also would like to know the computational time, see more on the question section (**)

From this viewpoint, I would give 6.


**Questions:**

- How much do the things need to be different in order to learn?

By learning something, how much the SORL can distinguish some similar things?
If we do not know anything about the green apple, can we increase the chance to distinguish the different variety of the green apple by learning green signals? Maybe the difference between the variety of the green apple is embedded in the spectral gap, but do you have any insights on this?

- I'm just curious how did you come up with Eq. (6)?

- This is not the question but a comment connecting to (*) in the strength section.

The Eq.(5) gives the SVD, but in this case Eq.(5) gives the eigendecomposition. Since the normalized Laplacian $L$ is defined as L := I -A, the Eq. (5) gives the k smallest eigenvalues. Thus, since the $k$ smallest eigendecompoition of the normalized Laplacian corresponds to the relaxed solution of the normalized graph cut, the SORL is actually corresponds to the relaxed solution of the normalized graph cut (see Sec.5.3 in [i]).
From this viewpoint, as far as I understand, Thm 3.1 can be more powerful for the spectral clustering community -- the Eq. (6) corresponds the result of the spectral clustering (i.e., the relaxed solution of the normalized graph cut).

With this, the proposed method is not surprising (in a good sense); the proposed SORL is an open-world learning hinted normalized graph cut.

From this view, I'd like to connect the point (**), if this method is faster than SVD, this indicates that there can be some faster way of the spectral clustering?

- This is also minor but the authors might want to cite [i], which is the one of the most cited paper in spectral clustering.

[i] Von Luxburg, Ulrike. "A tutorial on spectral clustering." Statistics and computing 17 (2007): 395-416.



----

POST REBUTTAL

I'm satisfied with the authors' rebuttal, therefore I increase my score from 6 to 7.

---

> ### Author Rebuttal · Authors · 2023-08-08
>
> We thank the reviewer for the insightful questions! Below we address each of your comments in detail.
>
> #### **- Experiments with different labeling ratios.**
>
> Great suggestions! We provide the comparison by reducing the labeling ratio from 50% (default) to 25%, 10% and 0% on CIFAR-100, while keeping the number of known classes to be the same (i.e., 50 in CIFAR-100). The results suggest that a small portion of labels can lead to a strong improvement in the accuracy for both known and novel classes.
>
> | Labeling Ratio | Method |	All | Novel |	Seen |
> |--------|--------|----------|-------|-------|
> | 50% | SORL | 55.8 | 51.7 | 68.4 |
> | 25% | SORL | 54.4 | 50.5 | 64.4|
> | 10% | SORL | 51.6 | 50.4 | 60.2|
> | 0% (unsupervised) | SORL  | 37.3 | 40.2 | 38.3 |
>
> #### **- Insights on learning the variety of similar samples.**
> Sure! We are pleased to share insights on the subject of distinguishing similar samples, which revolves around three sub-questions.
> 1. *How much can SORL distinguish similar things?*
> The ability to differentiate between similar items using the SORL algorithm depends on the embedding size $k$, as determined by the neural network. Specifically, the size $k$ correlates to the rank of the approximation matrix used within the SORL algorithm. A larger $k$ provides a greater capacity to perceive subtle differences among similar samples. In our experiments on the CIFAR dataset, we set $k=1000$ to allow the encoding of a broad spectrum of variations, going beyond mere class distinctions.
>
> 2. *How to increase the chance to distinguish the different varieties?*
>  a) Should the learning capacity permit, increasing the dimensionality $k$ can extend the "volume" to encompass more varieties.
> b) Enhancing the augmentation strategy contributes to a more precise adjacency matrix, assisting the model in better discerning subtle differences between varieties.
>
> 3. *What is the relationship between the variety and the spectral gap?*
> At a high level, the magnitude of singular values ($\sigma$) reflects the importance of a specific variety (represented by a singular vector). A larger spectral gap implies a significant variation in importance levels. For instance, the $\sigma$ for features that distinguish red from green would be considerably larger than the $\sigma$ that differentiates light green from dark green.
>
> #### **- Rationale in deriving Eq. (6).**
> We derive Eq. (6) by unfolding the Frobenius norm of $\mathcal{L}_{mf}$. A high-level explanation of proof is in L153 to L155.
>
> #### **- Application of theorem to a broader spectral clustering community.**
> We are grateful for the reviewer's insightful comments and agree with the potential of applying our algorithm to traditional spectral clustering. In response to the points raised by the reviewer, we offer detailed discussions on the following subjects:
>
> + *Set up for learning tabular data.* Our SORL algorithm is constructed using an augmentation graph where each element represents the probability of considering two samples as positive pairs. The augmentation strategy can be tailored for tabular data; for example, by adding noise to continuous data or permuting items in discrete data. This approach essentially resembles distance-based graphs, such as the k-nearest neighbor graph.
> + *Applying the main theorem to the spectral clustering community*. Definitely! We can directly apply Theorem 4.1 (assuming the reference to 3.1 is a typo in the comments) to spectral clustering, provided it is built upon the augmentation graph as discussed in point (a).
> + *Computational aspects.* We concur with the reviewer's observation that SORL can be more efficient than running a full SVD on the entire graph. This efficiency is due to SORL's utilization of a sampling strategy. It's noteworthy to mention that while SORL is confined to the augmentation graph, SVD can be applied to all graphs.
> + *Citing [i]*. Thanks for pointing it out! We have added it to our revised draft.

---

> > ### Comment · Reviewer_7sHp · 2023-08-10
> > **Increasing my score from 6 to 7**
> >
> > Thank you very much for the rebuttal. I am satisfied with the additional experiments. I think that the experimental results make sense. I'd like to increase my score from 6 to 7.
> >
> > By camera-ready (you don't have to do in the rebuttal period in my opinion), I'm curious for the other split for the other methods, ideally one from open world based (e.g., ORCA) and one from non-open world (e.g., FixMatch), if this makes sense. Sometimes unsupervised is not possible, but I believe that probably you can use only one instance instead.
> >
> > The reason why I propose such experiment is that the SORL does not seem to grow fast in "novel," which is an important category for the open world learning setting as far as I understand.
> > From 0% to 10% the accuracy seems to grow fast, but from 10 to 25% accuracy does not change, to 50% the accuracy increase seemingly marginally. In contrast, "seen" seems to grow, which makes sense. But we can only judge this by comparing with others. If you figure out that the SORL is "slow" for the novel by doing the additional experiments, do you have any insights for slowness from the theory perspective? Is 10% all you need for novel? I expect to have some answers for this by camera-ready.
> >
> > Even if your method is somewhat weaker than the other methods in some sense from this perspective, I am assured your method is promising since your method almost beats the others at 50%.  I guess that the additional experiments will not reduce the value of this study and therefore I request this by the camera-ready, only if this makes sense.
> > At the same time, I understand that the additional experiments for the other methods may not be possible since the other results may be brought from the other papers.
> >
> > Thank you very much for the other answers for my questions. All of these make sense. I enjoyed reading the rebuttal as well as the manuscript.
> >
> > ---
> >
> > EDIT: At this point I do not know how to update the score in my main rebuttal as edit button disappears. Thus the visible rating remains 6. But don't worry, my rating in my mind at this point (as of 10th Aug) is 7. I'll update as soon as I figure out how to do so.

---

### Official Review · Reviewer_iP9L · 2023-07-07

**Soundness:** 4 excellent
**Presentation:** 4 excellent
**Contribution:** 4 excellent
**Rating:** 7
**Confidence:** 3

**Summary:**

The paper tackles the domain of open-world representation learning, which aims to learn representations that can correctly cluster samples in the novel class and classify samples in the known classes by utilizing knowledge from the labeled data. Notably, the motivation of this paper is to provide a theoretical foundation in this area by introducing a novel graph-theoretic framework specifically designed for open-world settings. Theoretically, the crux of the framework involves characterizing clustering through graph factorization. A new algorithm called Spectral Open-world Representation Learning (SORL) is introduced, which essentially involves spectral decomposition on the graph. The authors provide theoretical guarantees by deriving a provable error bound on the clustering performance for both known and novel classes. Practically, the paper also offers empirical evidence, demonstrating that SORL is competitive with, or even superior to, existing methods on benchmark datasets.

**Strengths:**

The paper is well-written with clear motivation and structure.
Overall interesting problem; good mathematical exposition; solid theoretical results and analysis; interesting and good experimental results

**Weaknesses:**

1. The experiments are not sufficient. Other large commonly used benchmark dataset is not used, like ImageNet-100.
2. The paper has a very good theoretical analysis. But it is still unclear what contributes to the performance improvement.

**Questions:**

1. It would be interesting to see the results in a larger benchmark dataset, ImageNet 100.

2. What is the effect of using different known and novel classes ratio besides 50%? Like, 0.25, 0.1.

3. How do you choose the hyper-parameters, $\eta_{u}$, $\eta_{l}$?

4. In the supplementary material, it is mentioned that pre-training is used. I wonder why this pre-training is necessary for this method and what it will be without pertaining.

**Limitations:**

The broader impact has been discussed. It has a positive potential impact on the community.

---

> ### Author Rebuttal · Authors · 2023-08-08
>
>
> We thank the reviewer for the insightful questions! Below we address each of your comments in detail.
>
> #### **- Results on ImageNet-100.**
>
> Sure! As suggested, we report below results on the ImageNet-100 benchmark, which is commonly compared in the literature.  For ImageNet, the training and evaluation setting is consistent with prior works (ORCA, GCD, OpenCon, etc), which have a division of 50 known classes and 50 novel classes.
>
>
> The empirical results suggest that SORL is comparable to the current state-of-the-art methods with a <1% gap in terms of overall accuracy. Beyond chasing the benchmarks, the significance of our work lies in providing an in-depth theoretical understanding of the open-world representation learning task (which is urgently lacking in the field).
>
>
> | Method |	All | 	Novel |	Seen |
> |--------|----------|-------|-------|
> | FixMatch  (Kurakin et al., 2020) |  34.9 | 36.7 | 65.8 |
> | DSL	(Guo et al., 2020)  |  30.8 | 32.5 | 71.2 |
> | CGDL (Sun et al., 2020) |  31.9 | 33.8 | 67.3 |
> | DTC	(Han et al., 2019) |  21.3 | 20.8 | 25.6 |
> | RankStats  (Zhao & Han, 2021) 	|  40.3 | 28.7 | 47.3 |
> | SimCLR  (Chen et al., 2020a)  |  36.9 | 35.7 | 39.5 |
> | ORCA (Cao et al., 2022) |  76.4 | 68.9 | 89.1 |
> | GCD	 (Vaze et al., 2022) |  75.5 | 72.8 | 90.9 |
> | OpenCon  (Sun et al. 2023)	|  83.8 | 80.8 | 90.6 |
> | SORL (Ours)	|  82.9 | 80.4 | 89.2 |
>
>
> #### **- Discussion on the performance improvement over baseline methods.**
>
> We are happy to share insights on the differences in the representation learning strategies that have been implemented in baseline methods. Two of the most recent methods, GCD and OpenCon, employ a combinatory training approach that leverages both supervised contrastive learning (SupCon) with labeled samples, and self-supervised contrastive learning (SimCLR) with unlabeled samples. A detailed theoretical explanation of this strategy is available in Appendix D.  At a high level, the feature space derived from “SupCon+SimCLR” is given by the top-k eigenvectors of a matrix different from SORL. This insight serves as an invitation to other researchers to further explore and provide more comprehensive comparisons between different representation learning strategies.
>
> #### **- Results with different known/novel class ratios.**
>
> Great suggestions! We provide a comparison by reducing the known class number from 50 (default) to 25 and 10 on CIFAR-100, while keeping the labeling ratio to be the same (i.e., 50%). The results suggest that using more known classes will help the clustering performance of the novel classes.
> | Known Classes Number | Method |	All | Novel | Seen |
> |--------|--------|----------|-------|-------|
> | 50 | SORL | 55.8 | 51.7 | 68.4 |
> | 25 | SORL | 50.1 | 47.7 | 68.5 |
> | 10 | SORL | 48.6 | 47.3 | 66.0 |
>
> #### **- Strategy in choosing hyperparameters $\eta_u$ and $\eta_l$.**
>
> Good question! We follow the validation strategy proposed in prior works of open-world representation learning (See Appendix I in Sun et al., 2023). Specifically, we split the classes in $\mathcal{Y}_l$ equally into two parts: known classes and “novel” classes (for which we know the labels). Moreover, 50% of samples in the selected known classes are labeled. The constructed validation dataset is used to select the hyper-parameters by grid searching.
>
> #### **- The necessity of using a pre-trained model.**
>
> Fair concern! We use the pre-trained model mainly for several reasons:
>
> 1. *Adhering to Protocols in Prior Works*. The learning setting that encompasses both labeled and unlabeled data, along with a mixture of known and novel classes, was first introduced in ORCA (Cao et al., 2022). This approach employs the ORCA algorithm on a ResNet model pre-trained using self-supervised loss, a strategy also applied in subsequent works (such as OpenCon, GCD, etc.). To maintain a fair comparison, we adhere to this established convention.
>
> 2. *Efficiency in Training*. Importantly, by utilizing a pre-trained backbone, both prior works and SORL focus on training only the projection layers and the last block (layer4) of the ResNet, while keeping the preceding blocks fixed. This approach is more effective in training by consuming less GPU memory and significantly reducing the number of gradient update steps.
>
> 3. *Emulating the Real-World Pipeline*. In the industry, it's standard practice to first train a model using self-supervised loss on a large corpus of unlabeled data. Leveraging such a pre-trained model enables a wide array of downstream tasks depending on the additional labeled information available. The exploration of the label perturbation on the pre-trained model constitutes a key research question investigated in this paper.
>
> What if we do not apply the pertaining? As we have shown in the following table, they exhibit similar performance.
>
> | Pretrain Epochs | Training Epochs | Method | All | 	Novel |	Seen |
> |--------|--------|----------|-------|-------|-------|
> | 1200 | 400 | SORL | 55.8 | 51.7 | 68.4 |
> | 0 | 1600  | SORL | 55.4 | 52.2 | 67.8 |

---

> > ### Comment · Reviewer_iP9L · 2023-08-20
> >
> > Thanks for your responses. Having read the rebuttal, I will increase my initial score.

---

> > > ### Author Response · Authors · 2023-08-20
> > > **Thank you!**
> > >
> > > We would like to thank the reviewer for taking the time to read our rebuttal and for your positive feedback!
> > >
> > > Best,
> > >
> > > Authors

---

### Author Rebuttal · Authors · 2023-08-08


We thank all the reviewers for their constructive and valuable feedback.

We are honored that the reviewers acknowledge the novelty of our graph-theoretic framework (R1, R3, R4) with excellent contribution (R1, R4) and soundness (R1).
Multiple reviewers value the theoretical nature of our paper (R4), finding theoretical insight to be interesting (R2) and solid (R1). Beyond theoretical insight, the reviewers recognized the practical values of the framework (R4), with the approach to be simple and effective (R3) and empirical results outperforming (R1, R2, R3) the existing methods in the open-world learning setting. We are equally glad with R2's enjoyment in reading the paper, and the comments with excellent presentation (R1, R3), well-written (R1, R4), clear motivation and structure (R1) of our paper.

We have addressed the reviewers’ comments and concerns in individual responses to each reviewer.

(\* As abbreviations, we refer to **Reviewer iP9L** as R1, **Reviewer 7sHp** as R2, **Reviewer 66eW** as R3, and **Reviewer 1J6p** as R4 respectively.)

As suggested by R4, we added the following paragraphs to the draft:

+ Added the paragraph “Analysis goal” in Section 2:

*Our analysis goal is to comprehend the role of label information in shaping representations for both known and novel classes. it's essential to note that our theoretical approach aims to understand the perturbation in the clustering performance by labeling existing, previously unlabelled data points within the dataset.  By contrasting the clustering performance before and after labeling these instances, we uncover the underlying structure and relations that the labels may reveal. This analysis provides invaluable insights into how labeling information can be effectively leveraged to enhance the differentiation of both known and novel classes.*
+ Added the limitation discussion section.

*Limitation: Our theoretical framework has two potential limitations to be considered in practice: a)  The augmentation graph serves as a potent theoretical tool for elucidating the success of modern representation learning methods. However, it is challenging to ensure that current augmentation strategies, such as cropping, color jittering, or Gaussian blurring, can transform two dissimilar images into identical ones.  b) The utilization of Theorems 4.1 and 4.2 necessitates an explicit knowledge of the adjacency matrix of the augmentation graph, a requirement that can be intractable in practice.
In light of these limitations, we encourage further research to enhance the practicality of these theoretical findings.*

---

> ### Comment · Reviewer_1J6p · 2023-08-11
>
> Thank you for adding these two helpful paragraphs!

---

### Decision · Program_Chairs · 2023-09-21

**Decision:**

Accept (spotlight)

**Comment:**

The paper aims at improving the theoretical understanding of open-world representation learning by introducing a novel graph-theoretic framework. The paper is well written and the mathematical exposition is clear. Although there were some concerns regarding the breadth of the numerical examples, the authors have addressed those in the rebuttal to the satisfaction of the reviewers.